# The lncRNA *Caren* antagonizes heart failure by inactivating DNA damage response and activating mitochondrial biogenesis

Michio Sato [1,2,3,15], Tsuyoshi Kadomatsu [1,4,15], Keishi Miyata[1,4,5,15], Junco S. Warren[3,6,7], Zhe Tian[1], Shunshun Zhu[1], Haruki Horiguchi[1,8], Aman Makaju[6], Anna Bakhtina [6], Jun Morinaga[1], Taichi Sugizaki[1], Kaname Hirashima[1], Kumiko Yoshinobu[9], Mai Imasaka[10], Masatake Araki [9], Yoshihiro Komohara[11], Tomohiko Wakayama[12], Shinichi Nakagawa[13], Sarah Franklin[6,7,14], Koichi Node [2], Kimi Araki [4,10] & Yuichi Oike [1,4,8 ✉]

In the past decade, many long noncoding RNAs (lncRNAs) have been identified and their in vitro functions defined, although in some cases their functions in vivo remain less clear. Moreover, unlike nuclear lncRNAs, the roles of cytoplasmic lncRNAs are less defined. Here, using a gene trapping approach in mouse embryonic stem cells, we identify *Caren* (short for cardiomyocyte-enriched noncoding transcript), a cytoplasmic lncRNA abundantly expressed in cardiomyocytes. *Caren* maintains cardiac function under pathological stress by inactivating the ataxia telangiectasia mutated (ATM)-DNA damage response (DDR) pathway and activating mitochondrial bioenergetics. The presence of *Caren* transcripts does not alter expression of nearby (*cis*) genes but rather decreases translation of an mRNA transcribed from a distant gene encoding histidine triad nucleotide-binding protein 1 (Hint1), which activates the ATM-DDR pathway and reduces mitochondrial respiratory capacity in cardiomyocytes. Therefore, the cytoplasmic lncRNA *Caren* functions in cardioprotection by regulating translation of a distant gene and maintaining cardiomyocyte homeostasis.

[1] Department of Molecular Genetics, Graduate School of Medical Sciences, Kumamoto University, Kumamoto, Japan. [2] Department of Cardiovascular Medicine, School of Medicine, Saga University, Saga, Japan. [3] Division of Kumamoto Mouse Clinic (KMC), Institute of Resource Developmental and Analysis (IRDA), Kumamoto University, Kumamoto, Japan. [4] Center for Metabolic Regulation of Healthy Aging (CMHA), Graduate School of Medical Sciences, Kumamoto University, Kumamoto, Japan. [5] Department of Immunity, Allergy, and Vascular Biology, Graduate School of Medical Sciences, Kumamoto University, Kumamoto, Japan. [6] Nora Eccles Harrison Cardiovascular Research and Training Institute, University of Utah, Salt Lake City, UT, USA. [7] Department of Internal Medicine, University of Utah School of Medicine, Salt Lake City, UT, USA. [8] Department of Aging and Geriatric Medicine, Graduate School of Medical Sciences, Kumamoto University, Kumamoto, Japan. [9] Division of Bioinformatics, Institute of Resource Developmental and Analysis (IRDA), Kumamoto University, Kumamoto, Japan. [10] Division of Developmental Genetics, Institute of Resource Developmental and Analysis (IRDA), Kumamoto University, Kumamoto, Japan. [11] Department of Cell Pathology, Graduate School of Medical Sciences, Kumamoto University, Kumamoto, Japan. [12] Department of Histology, Graduate School of Medical Sciences, Kumamoto University, Kumamoto, Japan. [13] RNA Biology Laboratory, Faculty of Pharmaceutical Sciences, Hokkaido University, Sapporo, Japan. [14] Department of Biochemistry, University of Utah, Salt Lake City, UT, USA. [15]These authors contributed equally: Michio Sato, Tsuyoshi Kadomatsu, Keishi Miyata. ✉email: oike@gpo.kumamoto-u.ac.jp

The protein-coding genome constitutes almost the entire genome of unicellular yeast[1]; by contrast, <1% of the human genome harbors protein-coding gene sequences and much of the rest is transcribed as non-coding RNA (ncRNA)[2], suggesting that the noncoding transcriptome is critical for the complexity of higher eukaryotes. In the past decade, many long ncRNAs (lncRNAs) have been identified and their in vitro functions defined[3–5], whereas the in vivo functions in physiological or pathological contexts are not yet fully clarified. Moreover, at present, many nuclear lncRNAs are known to regulate transcription of nearby (cis) genes through specific cis-regulatory sequences[6–9], while the roles of cytoplasmic lncRNAs remain less defined. Unlike messenger RNAs (mRNAs), lncRNAs cannot be functionally characterized based on sequence alone[10]. We previously used a reporter-based gene trapping strategy in embryonic stem (ES) cells to identify uncharacterized genes functioning in developmental and pathophysiological states[11,12]. That strategy generates random, sequence-tagged insertional mutations and can be coupled to expression and/or function-based assays. We have applied this approach to identify and characterize functional cytoplasmic lncRNAs.

DNA damage and subsequent activation of the DNA damage response (DDR) occurs in cardiomyocytes of heart failure (HF) patients[13–15]. Moreover, recent studies suggest that DNA damage is induced in cardiomyocytes during transverse aortic constriction (TAC)-induced heart failure development[16,17]. Increased production of reactive oxygen species (ROS) due to mitochondrial dysfunction is seen in the TAC heart, and ROS scavenging reportedly prevents TAC-induced HF[18,19]. Thus, oxidative stress following excess ROS production from mitochondria could underlie TAC-induced DNA damage in cardiomyocytes.

Here, we report that a lncRNA that we designate Caren is predominantly expressed in the heart and that Caren transcripts are localized to the cardiomyocyte cytoplasm. Interestingly, Caren transcript levels significantly decreased in the aging mouse heart and in mouse models of angiotensin II-induced cardiac hypertrophy or heart failure induced by TAC. Caren-deficient mice subjected to TAC showed accelerated HF relative to comparably treated wild-type (WT) littermates. Conversely, Caren overexpression in the mouse heart conferred TAC-induced HF resistance. Heart tissue overexpressing Caren transcripts showed activated mitochondrial biogenesis. Caren overexpression also blocked the upregulation of histidine triad nucleotide-binding protein 1 (Hint1), which activates ataxia telangiectasia mutated (ATM) serine/threonine kinase, a DNA damage response (DDR) regulator closely linked to HF development and progression[17,20]. Overall, we identify a cytoplasmic lncRNA that counteracts HF development by inactivating the ATM-DDR pathway and activating mitochondrial bioenergetics by regulating the translation of a distant (trans) gene.

## Results

**Identification of a long noncoding RNA expressed in cardiomyocytes**. To identify uncharacterized lncRNAs functioning in development, normal physiology, or pathophysiology, we conducted gene trapping analysis in ES cells using a trapping vector carrying a β-geo reporter gene[11,12]. As of December 8th, 2011, we had identified 1051 insertional mutant ES clones (http://egtc.jp/action/main/index). Using 5' rapid amplification of cDNA ends methodology[21], we identified trapped transcripts based on the insertion of the trapping vector into the genome of the 1051 clones. Database analysis indicated 912 transcripts of known protein-coding genes plus 139 uncharacterized transcripts (Fig. 1a). The trapping vector was evenly inserted into the genome of chromosomes (Supplementary Fig. 1a), except for the Y.

Among 139 uncharacterized transcripts, regions corresponding to 13 were located within intergenic regions known as K4-K36 domains (Supplementary Fig. 1b), which reportedly encode long intergenic ncRNAs (lincRNAs)[22]. We then generated chimeric mice for each of these 13 ES clones and mated them with C57BL/6N females to generate F1 mice. Overall, we succeeded in generating 13 mouse lines showing germ-line transmission of insertional mutations in functionally uncharacterized lncRNAs. We then performed an expression-based assay of day 12.5 postconception (dpc) heterozygous embryos based on β-geo reporter expression, which we detected by X-gal staining. In one line (T167) we observed abundant expression of the trapped lncRNA restricted to the embryonic heart (Fig. 1b). Given that similarly expressed genes, such as Nppa, Nppb, and Myh7, function in cardiac hypertrophy and HF[23–25], we further analyzed the lncRNA trapped in this line.

Genomic analysis revealed that the insertion occurred in the intron between exons 2 and 3 of the 2500002B13Rik gene (Fig. 1c). Restriction mapping and sequencing analyses showed deletion of 14 bp of genomic DNA at the insertion. We confirmed that 2500002B13Rik resides between K4-K36 domains using ENCODE data[26,27] (Fig. 1d). X-gal staining of cardiac tissues from 10-week-old T167 mice also revealed abundant 2500002B13Rik expression in adult mouse hearts (Fig. 1e). Adult mice also showed 2500002B13Rik expression in various tissues, such as kidney and skeletal muscle, although expression was highest in the heart (Supplementary Fig. 1c). Moreover, in αMHC-EGFP Tg mice, we detected 2500002B13Rik transcripts in GFP-positive cardiomyocytes, while GFP-negative non-cardiomyocytes did not express 2500002B13Rik transcripts (Fig. 1f and Supplementary Fig. 1d). Thus, we designated the corresponding gene, "Cardiomyocyte-enriched transcript (Caren)". To confirm that Caren is a lncRNA, we conducted in silico analysis using the Coding Potential Calculator (http://www.cpc.cbi.pku.edu.cn)[28] (Fig. 1g) and found that the protein-coding score of Caren RNA sequences was lower than that of known lncRNAs, such as Xist, Bvht, Fendrr, and Mhrt. RT-PCR analysis also revealed that in contrast to the lncRNA Hotair, which is reportedly nuclear[29–31], Caren transcripts were primarily localized to the cardiomyocyte cytoplasm and were absent from the nucleus (Fig. 1h).

In mice, Caren resides on chromosome 8 between the high mobility group box 2 (Hmgb2) and Sin3A-associated protein 30 (Sap30) genes, the respective 3'-neighboring and 5'-neighboring genes (Fig. 1d). Human HMGB2 and SAP30 are similarly arrayed on human chromosome 4 (Supplementary Fig. 2a). Moreover, analysis of a human genomic DNA database (GRCh38/hg38 data) revealed that the intergenic region between human HMGB2 and SAP30 gives rise to 6 lincRNA transcripts (Supplementary Fig. 2b). Two of them (transcripts 1 and 3) were expressed in human inducible pluripotent stem cell-derived cardiomyocytes (iPSC-CMs) (Supplementary Fig. 2b).

**Caren deficiency accelerates cardiac dysfunction in HF model mice**. Caren transcript levels in the heart of aged (24-months-old) mice were significantly lower than those in young (2-months-old) mice (Fig. 1i, left). Moreover, in mice, Caren transcript levels were significantly lower in the hypertrophic heart induced by angiotensin II (Ang II) infusion relative to vehicle-infused control heart (Fig. 1i, middle). When we assessed the failing mouse heart induced by pressure overload caused by transverse aorta constriction (TAC)[32,33], Caren transcript levels were significantly lower in the TAC relative to sham-operated control heart (Fig. 1i, right). To assess Caren loss-of-function, we analyzed mice homozygous for T167 (Caren^{β-geo/β-geo}) (Fig. 1c and Supplementary Fig. 3a). Homozygotes were born alive in

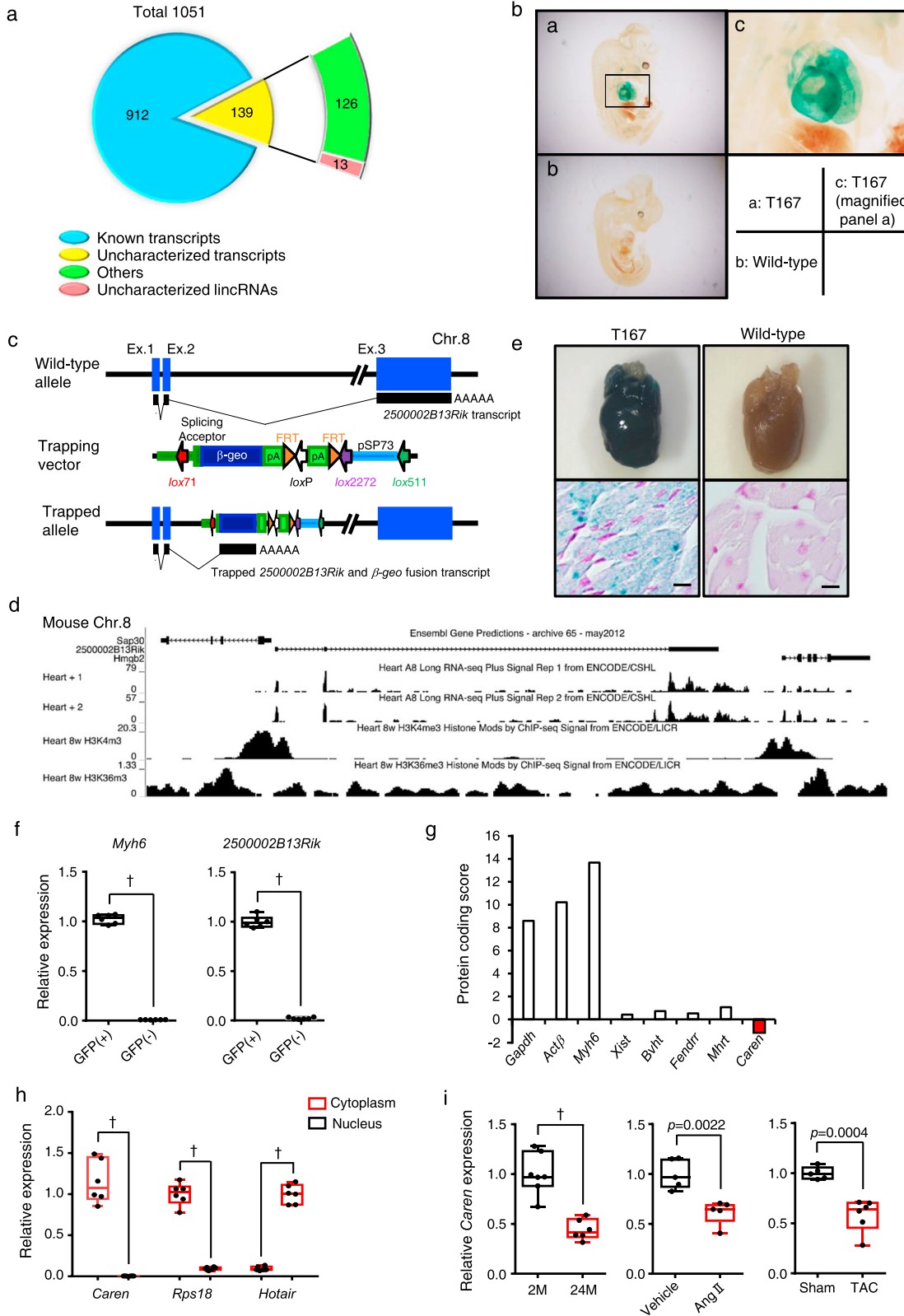

accordance with Mendelian inheritance (Supplementary Fig. 3b), and 12-week-old $Caren^{\beta-geo/\beta-geo}$ mice showed no $Caren$ transcripts in heart tissue (Supplementary Fig. 3c), suggesting that $Caren$ is not essential for cardiac development. To determine whether $Caren$ functions in myocardial remodeling under pressure overload, 12-week-old $Caren^{\beta-geo/\beta-geo}$ mice and WT littermates were subjected to TAC surgery. Four weeks later, WT mice developed adaptive cardiac

hypertrophy with moderate left ventricular dilatation (Fig. 2a–c and Supplementary Data 1). By contrast, $Caren^{\beta-geo/\beta-geo}$ mice developed marked left ventricular dilatation with a significant decrease in fractional shortening, resulting in HF development accompanied by lung congestion (Fig. 2a–c and Supplementary Data 1). Both heart weight/body weight (HW/BW) and lung weight/body weight (LW/BW) ratios significantly increased in $Caren^{\beta-geo/\beta-geo}$ relative to

**Fig. 1 Identification of a long noncoding RNA abundantly expressed in cardiomyocytes. a** Distribution of RNAs identified in gene trapping strategy in ES cells. **b** Expression of lacZ from targeting vector at mouse E12.5. Heart cells show X-gal positivity. **c** Structure of the WT allele, as well as the trapping vector pU21 and the *Caren* insertion site in mouse chromosome 8. **d** UCSC genome browser snapshot showing locus containing *2500002B13Rik* (*Caren*) and flanking *Hmgb2* and *Sap30* genes on chromosome 8 (NCBI37/mm9), and RNA-seq signals in adult mouse heart based on ENCODE/CSHL, as well as H3K4me3 and H3K36me3 ChIP-seq signals in mouse adult heart based on ENCODE/LICR. **e** X-gal staining in heterozygous T167 (*Caren*$^{+/\beta\text{-}geo}$) mice. LacZ expression was detected in the heart (top row, left) and in the myocardium of the left ventricle (bottom row, left). Sections were counterstained with nuclear fast red. Scale bar, 100 μm. **f** Quantitative RT-PCR analysis of *Myh6* and *2500002B13Rik* (*Caren*) expression in GFP$^+$ and GFP$^-$ cells isolated from heart tissues of α*MHC-EGFP* Tg mice (*n* = 6 per group). Values derived from GFP$^+$ cells were set to 1. **g** Protein coding potential scores of known mRNAs (*Gapdh*, *Actinβ*, and *Myh6*) or lncRNAs [*Xist*, Braveheart (*Bvht*), *Fendrr*, and *Mhrt*] compared to *Caren*. **h** Quantitative RT-PCR analysis of *Caren*, *Rps18* (a mature mRNA), and *Hotair* (a nuclear lncRNA) expression in nuclear and cytoplasmic fractions from the whole heart of 12-week-old wild-type mice (*n* = 6 per group). For *Caren* and *Rsp18*, cytoplasmic expression levels were set to 1. For *Hotair*, nuclear expression levels were set to 1. **i** Relative *Caren* transcript levels in hearts from 2 (2M)-month-old and 24 (24M)-month-old WT mice (2 M: *n* = 7, 24 M: *n* = 6) (left), from WT mice 2 weeks after Angiotensin II (Ang II) or vehicle treatment (*n* = 5 per group) (middle), and from WT mice 6 weeks after TAC or sham surgery (sham: *n* = 5, TAC: *n* = 6) (right). Levels seen in respective 2-month-old, vehicle-treated, or sham-operated mice were set to 1. Box plots for **f**, **h**, and **i** present min to max, median, and all points. Statistical significance was determined by a two-sided unpaired Student's *t*-test (**f**, **h**, **i**). $^{\dagger}p < 0.0001$, n.s; not significant, between genotypes or groups.

control WT mice (Fig. 2d). Moreover, cardiomyocyte enlargement, as estimated by histological analysis of heart tissue, was significantly greater in the *Caren*$^{\beta\text{-}geo/\beta\text{-}geo}$ TAC heart compared to the WT TAC heart (Fig. 2e, f). RT-PCR analysis of heart tissue showed that expression levels of markers of HF (*Nppa*, *Nppb*, and *Myh7*) and cardiac fibrosis (*Collagen1* and *Ctgf*) significantly increased in *Caren*$^{\beta\text{-}geo/\beta\text{-}geo}$ TAC relative to control TAC mice (Fig. 2g). By 140 days after TAC, more than 70% of *Caren*$^{\beta\text{-}geo/\beta\text{-}geo}$ mice had died, while only 20% of control mice succumbed (Fig. 2h). These results suggest that *Caren* deficiency increases vulnerability to pressure overload and aggravates cardiac dysfunction and HF-associated death.

**Mice overexpressing *Caren* resist HF development.** To determine whether *Caren* overexpression protects mice against TAC-induced HF development, we generated transgenic (Tg) mice overexpressing *Caren* driven by the CAG promoter (CAG-*Caren* Tg mice). To avoid additional insertional mutagenesis, we inserted *Caren* cDNA into the endogenous *Caren* locus on chromosome 8 of heterozygous T167 mice (*Caren*$^{+/\beta\text{-}geo}$) using the *Cre*/mutant *lox*/*loxP* recombination system[34] (Supplementary Fig. 4a). We confirmed that *Caren* expression in hearts of CAG-*Caren* Tg mice was 40 times higher than that seen in WT littermates (Supplementary Fig. 4b). Six weeks after TAC, WT littermate controls developed left ventricular dilatation with reduced fractional shortening, leading to HF development (Fig. 3a–c and Supplementary Data 1). Decreases in fractional shortening were milder in CAG-*Caren* Tg mice, whose hearts exhibited adaptive cardiac hypertrophy without ventricular dilatation (Fig. 3a–d and Supplementary Data 1). Furthermore, cardiomyocyte enlargement seen after pressure overload was attenuated in CAG-*Caren* Tg relative to control TAC mice (Fig. 3e, f). Moreover, expression levels of HF and cardiac fibrosis markers were significantly lower in CAG-*Caren* Tg compared with control mice (Fig. 3g).

To confirm that cardiomyocytes showing abundant *Caren* were resistant to HF, we generated α*MHC-Caren* Tg mice in which *Caren* expression was driven by the cardiomyocyte-specific α*MHC* promoter (Supplementary Fig. 5a, b). Six weeks after TAC, non-Tg littermate control mice developed ventricular dilation, concurrent with declined fractional shortening and cardiomyocyte enlargement, all of which were attenuated in α*MHC-Caren* Tg mice (Supplementary Fig. 5c–h and Supplementary Data 1). Moreover, expression levels of HF and cardiac fibrosis markers were also significantly lower in α*MHC-Caren* Tg compared with control mice (Supplementary Fig. 5i), suggesting that increased *Caren* transcript levels in cardiomyocytes are required for an anti-HF effect.

***Caren* regulates mitochondrial biogenesis in the heart.** Expression of the flanking *Hmgb2* and *Sap30* genes (see Fig. 1d) was relatively unchanged in hearts of both CAG-*Caren* Tg and *Caren*$^{\beta\text{-}geo/\beta\text{-}geo}$ mice (Supplementary Fig. 3d, e and Supplementary Fig. 4c, d), suggesting that phenotypes observed are not due to altered regulation of *cis* genes.

To determine mechanisms regulated by *Caren*, we performed LC-MS/MS-based proteomic analysis using whole tissue lysates of ventricles obtained from *Caren*$^{\beta\text{-}geo/\beta\text{-}geo}$, CAG-*Caren* Tg, and corresponding WT littermate mice. Enrichment analysis of differentially expressed proteins (118 of 922 proteins in *Caren*$^{\beta\text{-}geo/\beta\text{-}geo}$ mice, and 79 of 1076 in CAG-*Caren* Tg mice) was performed using STRING Database web-based tools[35]. The 10 most significantly enriched gene ontology (GO) terms for Kyoto Encyclopedia of Genes and Genomes (KEGG) pathways in *Caren*$^{\beta\text{-}geo/\beta\text{-}geo}$ and CAG-*Caren* Tg mice are shown in Fig. 4a. The most enriched term for both was metabolic pathways, including carbon metabolism, the tricarboxylic acid (TCA) cycle, and oxidative phosphorylation (OXPHOS). Accordingly, RT-PCR analysis showed significantly increased expression of genes functioning in mitochondrial biogenesis and OXPHOS (such as *Nrf2*, *Tfam*, *Cytc*, *Cox4*, and *Atp5a1*) in hearts of CAG-*Caren* Tg mice relative to WT littermates (Fig. 4b, c and Supplementary Fig. 4e). Moreover, the western blotting analysis showed that expression levels of protein components of mitochondrial electron transport chain complexes (I to V) consistently increased in CAG-*Caren* Tg relative to WT hearts (Fig. 4d). Notably, CAG-*Caren* Tg mice showed significantly increased TFAM protein levels in heart tissues compared with WT littermates (Fig. 4e). Furthermore, electron microscopy analysis revealed a greater number of mitochondria in heart tissue of CAG-*Caren* Tg mice than in WT littermate mice (Fig. 4f). Moreover, mitochondrial DNA content was significantly higher in CAG-*Caren* Tg relative to WT hearts (Fig. 4g). These findings suggest that *Caren* enhances mitochondrial biogenesis in the heart.

***Caren* deficiency decreases mitochondrial respiratory capacity in the heart.** We next asked whether *Caren* overexpression enhances mitochondrial respiratory function in cardiomyocytes. To do so, we performed extracellular flux analysis using mitochondria isolated from heart tissues of CAG-*Caren* Tg and WT littermate mice (Supplementary Fig. 6a–c). Basal respiration, phosphorylating respiration in the presence of ADP (state 3), resting respiration in the presence of oligomycin (state 4o), maximal uncoupling respiration in the presence of FCCP (state 3u), and respiration in the presence of antimycin A (Anti A) were comparable in CAG-*Caren* Tg and WT littermate

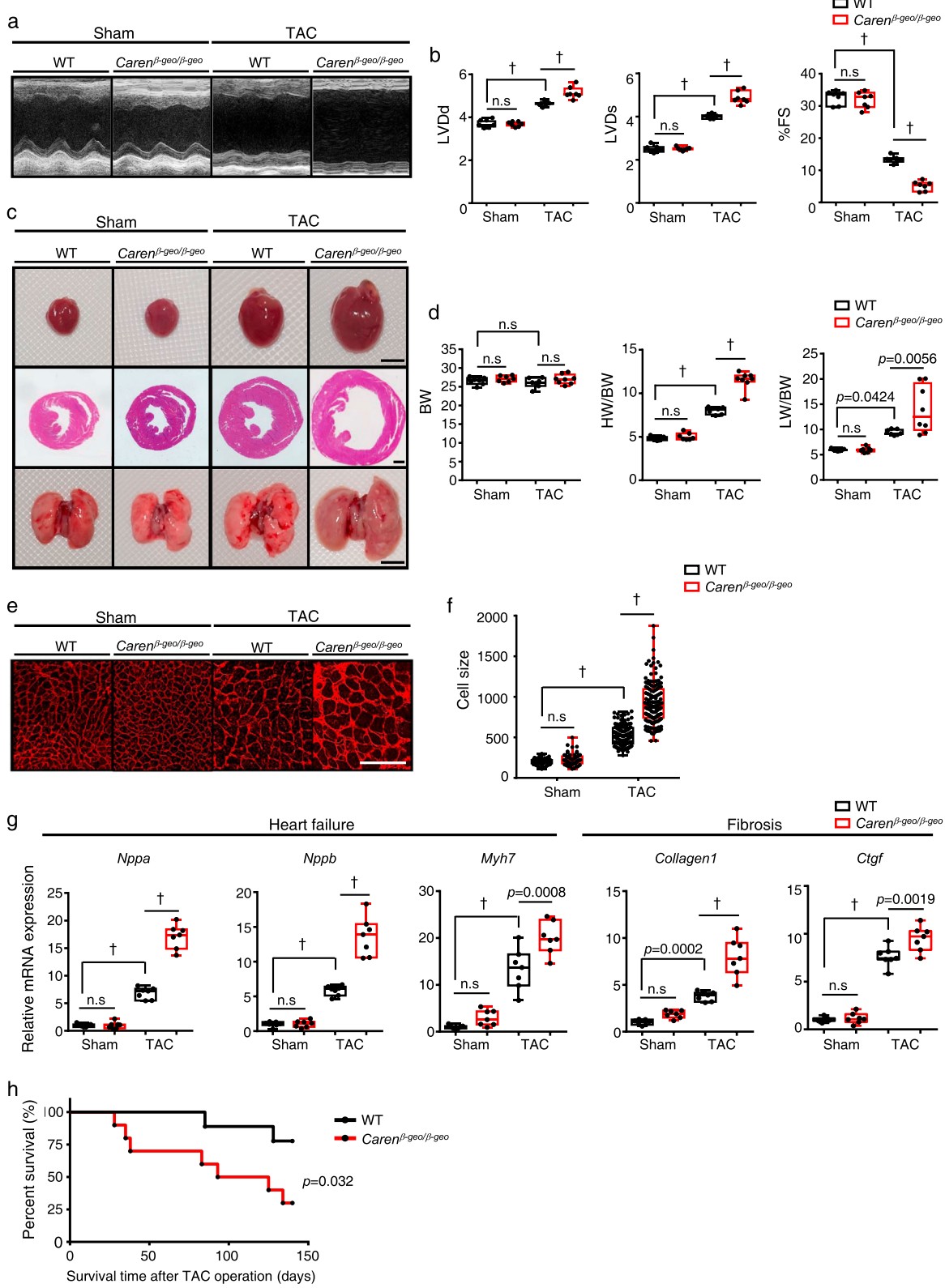

mice (Supplementary Fig. 6a, b). There was also no difference in the respiratory control ratio (RCR, state 3/state 4o) between CAG-*Caren* Tg and WT littermate mice (Supplementary Fig. 6c). BN-PAGE followed by immunoblotting analysis revealed comparable levels of mitochondrial respiratory chain complexes in the heart between genotypes (Supplementary Fig. 6d), and there was no difference in activities of those complexes

between CAG-*Caren* Tg and WT littermate mice (Supplementary Fig. 6e). These results suggest that *Caren* overexpression does not alter mitochondrial respiratory capacity at basal level but rather that the increase in protein components of mitochondrial respiratory chain complexes observed in hearts of CAG-*Caren* Tg mice is due to an increased number of mitochondria.

**Fig. 2 *Caren* deficiency accelerates cardiac dysfunction in a mouse HF model. a** Shown are representative M-mode echocardiography recordings in *Caren*$^{\beta\text{-}geo/\beta\text{-}geo}$ and WT littermate mice, 6 weeks after TAC or sham surgery. **b** Left ventricular end-diastolic diameter (LVDd) (mm) (left), left ventricular end-systolic diameter (LVDs) (mm) (middle) and percent fractional shortening (%FS) (right) in indicated mice ($n = 7$ per group). **c** Gross appearance of whole heart (top row; scale bar, 5 mm), hematoxylin-eosin (HE)-stained sections of the mid-portion of the heart (middle row; scale bar, 1 mm), and gross appearance of the whole lung (bottom row; scale bar, 5 mm). **d** Body weight (BW) (g) (left), HW/BW ratio (mg/g) (middle) and lung weight per body weight ratio (LW/BW) (mg/g) (right) in indicated mice (WT sham, *Caren*$^{\beta\text{-}geo/\beta\text{-}geo}$ sham, and WT TAC: $n = 7$ per group, *Caren*$^{\beta\text{-}geo/\beta\text{-}geo}$ TAC: $n = 8$). **e** Shown are representative left ventricle sections stained with wheat germ agglutinin (WGA) as an indicator of cardiomyocyte size. Scale bar, 100 μm. **f** Distribution of size of myocardial cells (μm$^2$) in indicated mice (WT sham: $n = 72$ cells, *Caren*$^{\beta\text{-}geo/\beta\text{-}geo}$ sham: $n = 58$ cells, WT TAC: $n = 128$ cells, *Caren*$^{\beta\text{-}geo/\beta\text{-}geo}$ TAC: $n = 136$ cells). This experiment was repeated independently three times with similar results. **g** Relative expression of genes associated with heart failure and fibrosis in hearts of indicated mice ($n = 7$ per group). Levels seen in sham-operated WT mice were set to 1. **h** Kaplan-Meier curve showing mortality after TAC ($n = 10$ per group). Box plots for **b**, **d**, **f**, and **g** present min to max, median, and all points. Statistical significance was determined by one-way ANOVA with Sidak's post hoc test (**b**, **d**, **f**, **g**) or log lank test (**h**). †$p < 0.0001$, n.s; not significant, between genotypes or groups.

In contrast to CAG-*Caren* Tg mice, we observed significantly decreased basal, state 3, and state 3u respiration in the heart mitochondria from *Caren*$^{\beta\text{-}geo/\beta\text{-}geo}$ mice compared to that from WT littermate mice (Supplementary Fig. 7a, b). The RCR in *Caren*$^{\beta\text{-}geo/\beta\text{-}geo}$ mice was also decreased relative to that in WT littermate mice, but that difference was not significant (Supplementary Fig. 7c). Although levels of mitochondrial respiratory chain complex I significantly increased in the heart of *Caren*$^{\beta\text{-}geo/\beta\text{-}geo}$ compared to WT littermate mice (Supplementary Fig. 7d), the activity of complex I significantly decreased in *Caren*$^{\beta\text{-}geo/\beta\text{-}geo}$ relative to WT littermate mice (Supplementary Fig. 7e). Moreover, *Caren*$^{\beta\text{-}geo/\beta\text{-}geo}$ mice exhibited a significant increase in levels of mitochondrial respiratory chain complex III in the heart relative to WT littermate mice (Supplementary Fig. 7d), while complex III activity was comparable between genotypes (Supplementary Fig. 7e). Taken together, these results suggest that *Caren* deficiency reduces mitochondrial respiratory capacity in cardiomyocytes.

**Caren suppresses translation of *Hint1* mRNA.** Proteomic analysis of proteins differentially expressed in *Caren*$^{\beta\text{-}geo/\beta\text{-}geo}$ relative to CAG-*Caren* Tg mice revealed the largest significant change to be increased levels of the tumor suppressor histidine triad nucleotide-binding protein 1 (Hint1) in the *Caren*$^{\beta\text{-}geo/\beta\text{-}geo}$ heart[36] (Fig. 5a). Consistently, western blotting analysis showed significant upregulation of Hint1 in the *Caren*$^{\beta\text{-}geo/\beta\text{-}geo}$ relative to the WT heart, while Hint1 protein levels significantly decreased in the CAG-*Caren* Tg compared to the WT heart (Fig. 5b, c). Interestingly, *Hint1* mRNA levels were comparable between genotypes (Fig. 5d), suggesting that *Caren* regulates Hint1 translation. To test this possibility, we assessed *Hint1* mRNA translation in vitro in the presence of sense or antisense *Caren* RNA. The presence of sense *Caren* significantly decreased Hint1 protein levels compared to controls in the assay, while the presence of antisense *Caren* had no effect in this context (Fig. 5e). Furthermore, the denatured form of *Caren* had no effect on *Hint1* mRNA translation (Supplementary Fig. 8). These results suggest that a normal secondary structure is required for *Caren* to suppress *Hint1* mRNA translation.

Given that *Caren* likely regulates *Hint1* mRNA translation, we hypothesized that *Caren* interacts with *Hint1* mRNA. To confirm this interaction, we performed an RNA pull-down assay (Fig. 5f). When we incubated either biotinylated *LacZ* mRNA (as a negative control) or biotinylated *Caren* transcripts with cytoplasmic lysates of mouse myoblast C2C12 cells, *Hint1* mRNA abundantly bound to *Caren* relative to the negative control. However, when biotinylated *Caren* RNA was incubated with total RNA from C2C12 cells, levels of *Hint1* mRNA bound to *Caren* markedly decreased, suggesting that *Caren* and *Hint1* mRNA interaction is indirect, as is the case with some lncRNAs that

reportedly regulate target mRNA translation by binding to RNA-binding proteins or ribosomes[37,38].

To further validate *Caren*/*Hint1* mRNA interaction, we performed RNA antisense purification (RAP) analysis using mouse embryonal carcinoma P19.CL6 cells crosslinked with disuccinimidyl glutarate (DSG) and formaldehyde (FA) (Fig. 5g). In RAP analysis using specific probes against *Hint1* mRNA (*Hint1* RAP), *Caren* transcripts co-precipitated with *Hint1* mRNA. By contrast, *Caren*/*Hint1* mRNA interaction was not observed in RAP analysis using specific probes against *LacZ* mRNA (*LacZ* RAP). Furthermore, *Caren*/*Hint1* mRNA interaction observed in *Hint1* RAP significantly decreased when we used cells that had not been crosslinked. Taken together, these results suggest that *Caren* may interact with *Hint1* mRNA through a protein intermediate(s).

**Increased Hint1 levels in cardiomyocytes impair mitochondrial respiratory function.** Our data suggest that *Caren* regulates cardiac energetics by suppressing Hint1 expression. We found that Hint1 protein levels gradually increase in the TAC heart as cardiac dysfunction progressed (Fig. 6a). To determine whether Hint1 modulates mitochondrial function, we established a cell line in which rat H9c2 cardiomyocytes overexpress HA-tagged mouse Hint1 (H9c2-Hint1) (Fig. 6b) in order to assess mitochondrial energetics. Mitochondrial membrane potential in H9c2-Hint1 cells significantly decreased compared with that seen in control cells (H9c2-mock) (Fig. 6c, d and Supplementary Fig. 9a, b). However, levels of reactive oxygen species (ROS) in mitochondria remained unchanged in H9c2-Hint1 relative to control cells (Fig. 6e and Supplementary Fig. 9c, d). To analyze mitochondrial respiratory capacity in cardiomyocytes overexpressing Hint1, we performed a Cell Mito Stress test using a Seahorse XF extracellular flux analyzer. We observed decreases in basal and maximal respiration levels and in ATP production, as well as decreased spare respiratory capacity in H9c2-Hint1 compared to control cells (Fig. 6f, g). These results suggest that the TAC-induced increase in Hint1 levels contributes to mitochondrial dysfunction in the failing heart, accelerating cardiac dysfunction.

**Hint1$^{+/-}$ mice resist HF development.** TAC-induced cardiac dysfunction and fibrosis were blocked in the CAG-*Caren* Tg heart in which Hint1 protein levels were ~35% those seen in WT littermates (Fig. 5c). To determine whether these phenotypes were due to decreased Hint1 protein levels, we generated *Hint1* knockout mice (Supplementary Fig. 10a–d) and subjected *Hint1* heterozygous knockout mice to TAC. Six weeks after surgery, littermate controls showed significantly decreased fractional shortening and ventricular dilation, as expected, whereas these changes were attenuated in *Hint1*$^{+/-}$ mice (Fig. 7a–d and Supplementary Data 1). Cardiomyocyte enlargement, as estimated by

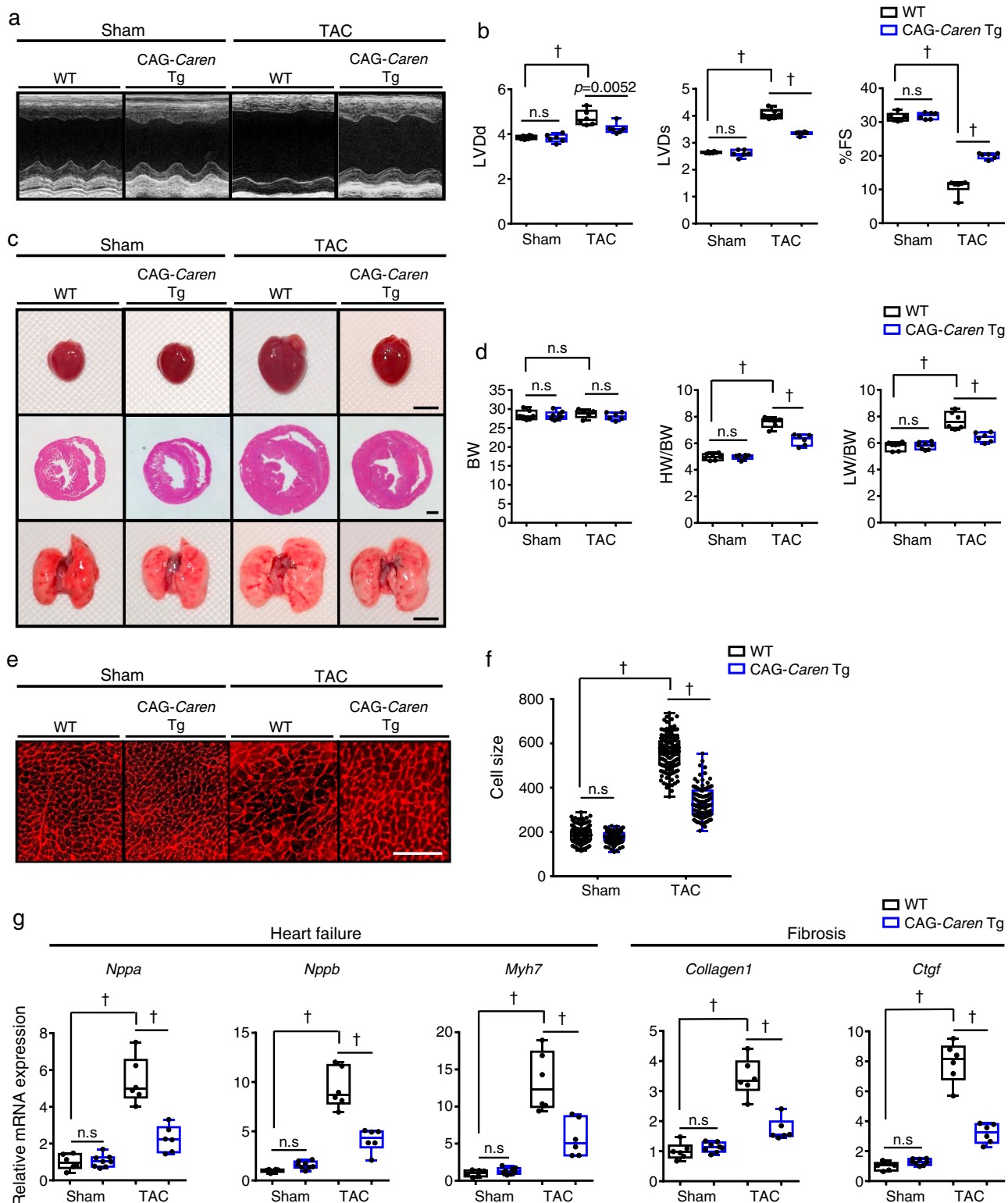

histological analysis of heart tissue, was also significantly decreased in *Hint1*[+/−] relative to control WT mice (Fig. 7e, f). Moreover, expression of HF and cardiac fibrosis markers were significantly lower in *Hint1*[+/−] relative to control mice (Fig. 7g). These anti-HF effects following TAC were similar to what we observed in CAG-*Caren* Tg mice (Fig. 3), strongly suggesting that *Caren* protects against HF by suppressing Hint1 expression.

However, in contrast to phenotypes seen in CAG-*Caren* Tg mice (Fig. 4), we did not observe significant differences in

expression levels of genes functioning in mitochondrial biogenesis and OXPHOS in heart tissue between *Hint1*[+/−] mice and WT littermates (Supplementary Fig. 10e–g), suggesting that Hint1-suppression under basal conditions does not alter mitochondrial biogenesis.

**Caren inhibits ATM activation during pressure overload by suppressing Hint1 expression.** In mouse embryonic fibroblasts,

**Fig. 3 Mice overexpressing *Caren* in cardiomyocytes resist HF development. a** Representative M-mode echocardiography recordings from CAG-*Caren* Tg and WT littermate mice, 6 weeks after TAC or sham surgery. **b** Left ventricular end-diastolic diameter (LVDd) (mm) (left), left ventricular end-systolic diameter (LVDs) (mm) (middle), and percent fractional shortening (%FS) (right) in indicated mice ($n = 6$ per group). **c** Gross appearance of whole heart (top row; scale bar, 5 mm), hematoxylin-eosin (HE)-stained sections of the mid-portion of the heart (middle row; scale bar, 1 mm), and gross appearance of the whole lung (bottom row; scale bar, 5 mm). **d** Body weight (BW) (g) (left), HW/BW ratio (mg/g) (middle), and lung weight per body weight ratio (LW/BW) (mg/g) (right) in indicated mice (WT sham: $n = 7$, CAG-*Caren* Tg sham: $n = 8$, WT TAC and CAG-*Caren* Tg TAC: $n = 6$ per group). **e** Shown are representative left ventricle sections stained with wheat germ agglutinin (WGA) to indicate cardiomyocyte size (scale bar, 100 μm). **f** Distribution of myocardial cells based on size (μm²) in indicated mice (WT sham: $n = 78$ cells, CAG-*Caren* Tg sham: $n = 131$ cells, WT TAC: $n = 50$ cells, CAG-*Caren* Tg TAC: $n = 102$ cells). **g** Relative expression of genes associated with heart failure and fibrosis in hearts of indicated mice (WT sham, WT TAC and CAG-*Caren* Tg TAC: $n = 6$ per group, CAG-*Caren* Tg sham: $n = 8$). Levels seen in sham-operated WT mice were set to 1. Box plots for **b**, **d**, **f**, and **g** present min to max, median, and all points. Statistical significance was determined by one-way ANOVA with Sidak's post hoc test (**b**, **d**, **f**, **g**). †$p < 0.0001$, n.s; not significant, between genotypes or groups.

Hint1 activity reportedly accelerates the DDR by activating ATM[20]. Moreover, a recent study showed that induction of DNA damage in mouse heart accelerates TAC-induced HF development, while $ATM^{+/-}$ mice are resistant to TAC[17]. Thus, TAC-induced Hint1 upregulation (Fig. 6a) may aggravate HF phenotypes during pressure overload by activating ATM and the subsequent DDR. To test this hypothesis, we asked whether TAC increased levels of ATM phosphorylation in the heart and also compared ATM phosphorylation in $Hint1^{+/-}$ vs. littermate WT TAC mice. ATM phosphorylation levels remarkably increased in the TAC heart, and those levels decreased in hearts of $Hint1^{+/-}$ TAC compared with WT TAC mice (Fig. 7h and Supplementary Fig. 11a). Moreover, ATM activation by TAC was significantly decreased in hearts of CAG-*Caren* Tg compared to WT control mice (Supplementary Fig. 11b). By contrast, ATM activation by TAC significantly increased in hearts of $Caren^{\beta\text{-}geo/\beta\text{-}geo}$ compared to WT control mice (Supplementary Fig. 11c).

We next investigated the significance of ATM activation in HF progression in TAC-operated $Caren^{\beta\text{-}geo/\beta\text{-}geo}$ mice. After TAC surgery, $Caren^{\beta\text{-}geo/\beta\text{-}geo}$ mice were intraperitoneally administered either the ATM inhibitor KU-60019 or PBS every 3 days for 6 weeks and cardiac function in each group was examined by ultrasonic echocardiography at various time points (Supplementary Fig. 12a). Relative to the PBS control group, left ventricular dilatation with reduced fractional shortening was partially, but significantly, suppressed in the ATM inhibitor-treated group (Supplementary Fig. 12b and Supplementary Data 1), as was the HW/BW ratio (Supplementary Fig. 12c). ATM activation was also markedly inhibited in the heart of ATM inhibitor-treated compared with control mice (Supplementary Fig. 12d). These results suggest that *Caren* counteracts HF development, in part, by preventing ATM-DDR pathway activation in response to pressure overload by suppressing Hint1 expression.

**Caren lncRNA structural integrity is important for anti-HF activity.** To identify a *Caren* sequence or domain required for the anti-HF effect, we divided full length (2618 nucleosides) *Caren* into 5 fragments of ~800 nucleosides each (fragments A, B, C, D, and E, from 5' to 3'), in which each fragment possessed 400 nucleosides of overlap with the adjacent fragment (Supplementary Fig. 13a). We then generated Tg mice expressing *Caren* fragments A–E, respectively, by inserting that fragment into the corresponding endogenous *Caren* genomic locus on chromosome 8 of T167 mouse heterozygotes ($Caren^{+/\beta\text{-}geo}$) using the *Cre*/ mutant *lox*/*loxP* recombination system (Supplementary Fig. 13b). None of these Tg mice exhibited an anti-HF effect against TAC (Supplementary Fig. 13c–g and Supplementary Data 1). We then assayed *Hint1* mRNA translation using full-length *Caren* or individual *Caren* Fragments (A-E) versus controls. That analysis indicated that full-length *Caren* significantly decreased Hint1 protein levels relative to controls; however, none of *Caren*

fragments altered *Hint1* mRNA translation (Supplementary Fig. 14). These results suggest that *Caren* anti-HF effects may require a structure best exhibited by full-length lncRNA rather than by a discrete *Caren* sequence.

**Re-expression of full-length Caren in failing heart slows HF progression in mice.** Next, we asked whether restoration of expression of full-length *Caren* in TAC-induced failing cardiomyocytes would improve cardiac function. To do so, we employed recombinant adeno-associated virus serotype 6 (AAV6) to specifically overexpress *Caren* in cardiomyocytes. We intravenously injected WT mice $1 \times 10^{11}$ vg per mouse of recombinant AAV6-*Caren* or $1 \times 10^{11}$ vg per mouse of recombinant AAV6-green fluorescent protein (GFP) as control virus. At 1 week after injection, the expression of *Caren* in hearts of mice injected with AAV6-*Caren* was 20 times higher than that seen in mice injected with AAV6-*GFP* (Fig. 8a). Moreover, we confirmed that mice injected with AAV6-*Caren* showed significantly decreased Hint1 protein levels, as well as increased expression of several genes functioning in mitochondrial biogenesis and OXPHOS in heart tissue compared to mice injected with AAV6-*GFP* (Fig. 8b and Supplementary Fig. 15a, b).

Next, we tested whether AAV6-mediated *Caren* overexpression would rescue cardiac hypertrophy and failure in response to TAC-induced pressure overload in WT mice. One week after TAC surgery, we confirmed that TAC-induced cardiac dysfunction was equivalent in all mice based on ultrasonic echocardiographic analysis (Fig. 8c, d). We then divided mice into three groups and intravenously injected animals in each group with $1 \times 10^{11}$ vg per mouse of recombinant AAV6-*Caren*, $1 \times 10^{11}$ vg per mouse of recombinant AAV6-*GFP*, or an equivalent dose of PBS as a control for venous injection, and examined cardiac function in each group by ultrasonic echocardiography at various time points (Fig. 8c and Supplementary Data 1). Both GFP and PBS control mice showed left ventricular dilatation with reduced fractional shortening, whereas mice treated with AAV6-*Caren* showed attenuated development of cardiac dysfunction and left ventricular dilatation (Fig. 8d–f and Supplementary Data 1). Both HW/BW and LW/BW ratios in mice injected with AAV6-*Caren* were similar to basal levels observed in sham-operated control mice (Fig. 9a, b). Cardiomyocyte enlargement was also attenuated in TAC mice injected with AAV6-*Caren* relative to both TAC control groups (Fig. 9c, d). Moreover, expression levels of HF and cardiac fibrosis markers were significantly lower in TAC mice injected with AAV6-*Caren* relative to TAC controls (Fig. 9e). ATM activation by TAC was also significantly decreased in hearts of TAC mice injected with AAV6-*Caren* compared to TAC mice injected with AAV6-*GFP* (Fig. 9f).

To confirm whether virally-based restoration of full-length *Caren* in failing heart rescues HF-associated death in $Caren^{\beta\text{-}geo/\beta\text{-}geo}$ mice, we analyzed survival of TAC-operated $Caren^{\beta\text{-}geo/\beta\text{-}geo}$ mice injected

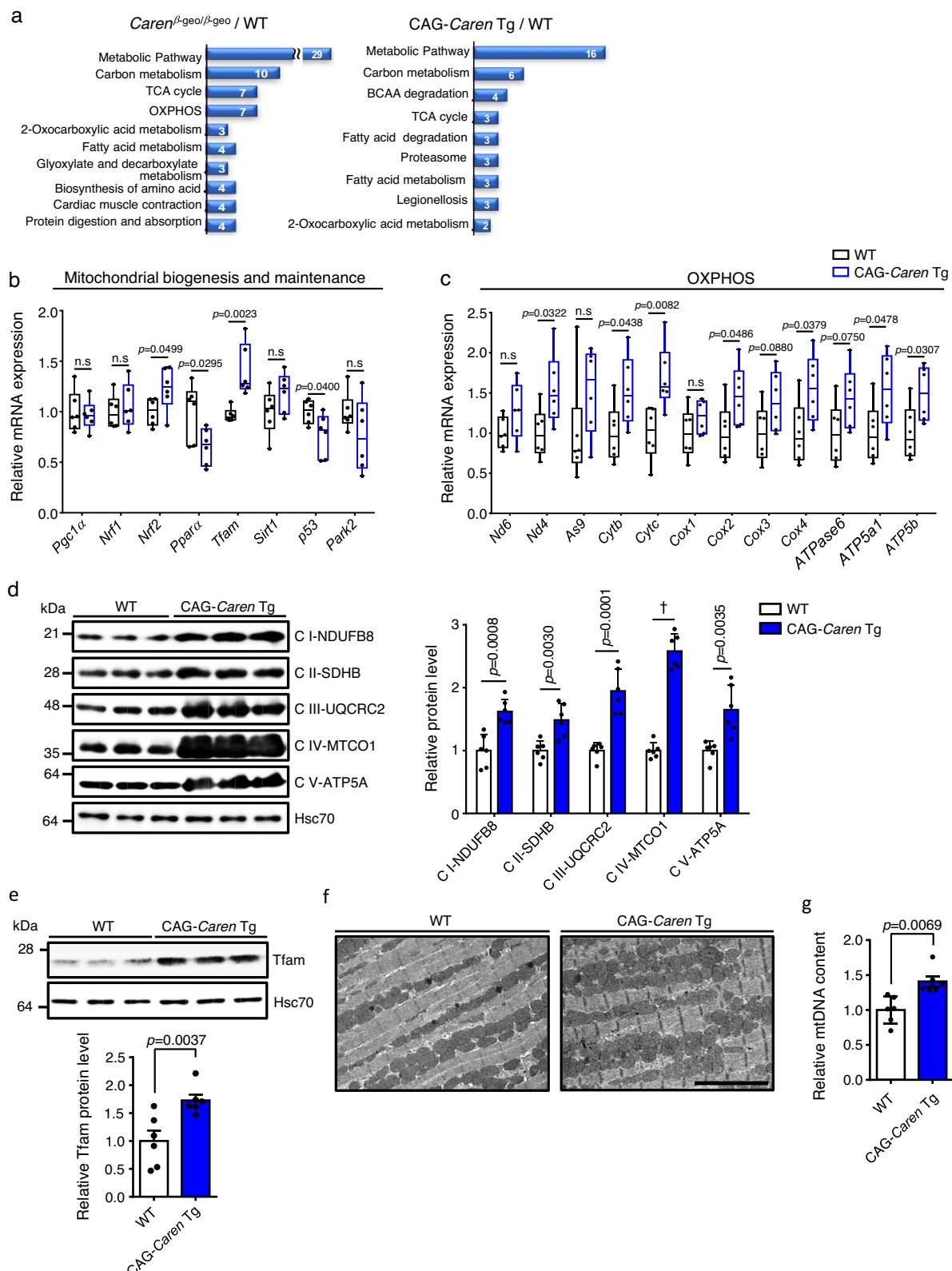

with AAV6-*Caren* or AAV6-*GFP*. By 220 days after TAC, all *Caren*^β-geo/β-geo mice injected with AAV6-*GFP* had died, while 50% of *Caren*^β-geo/β-geo mice that had received AAV6-*Caren* remained alive and showed significantly prolonged survival (Fig. 9g). These findings suggest that increased *Caren* transcript levels underlie *Caren* anti-HF effects.

**Analysis of candidate human *Caren* orthologues**. Finally, we asked whether either of the two lincRNA transcripts (1 or 3) seen in human cardiomyocytes (Supplementary Fig. 2b) was associated with cardiac function in humans. To do so, we examined the expression of both transcripts, as well as HINT1 protein in heart tissues obtained from autopsy cases from humans who died of

**Fig. 4 *Caren* overexpression enhances cardiac mitochondrial biogenesis. a** Enrichment analyses of significantly regulated proteins in heart tissues of 12-week-old *Caren*^β-geo/β-geo^ (left) and CAG-*Caren* Tg (right) mice for KEGG pathways ($n = 7$ each). **b**, **c** Relative expression of genes associated with mitochondrial biogenesis and maintenance (**b**) or mitochondrial oxidative phosphorylation (OXPHOS) (**c**) in heart tissues of 12-week-old CAG-*Caren* Tg and WT littermate mice ($n = 6$ per group). WT values were set to 1. **d** Representative western blot (left) and quantitation (right) of component proteins of specific mitochondrial electron transport chain complexes (I to V) in heart tissues of 12-week-old CAG-*Caren* Tg and WT littermate mice ($n = 6$ per group). Hsc70 served as a loading control. Protein intensity values in the WT group were set to 1. **e** Representative western blot (left) and quantitation (right) of TFAM protein levels in heart tissues of 12-week-old CAG-*Caren* Tg and WT littermate mice ($n = 6$ per group). Hsc70 served as a loading control. Protein intensity values in the WT group were set to 1. **f** Representative electron micrographs of heart tissue from WT (left) and *CAG-Caren* Tg (right) mice (scale bar, 5 μm). **g** Relative mitochondrial DNA (mtDNA) content in heart tissues of 12-week-old CAG-*Caren* Tg and WT littermate mice ($n = 6$ per group). WT values were set to 1. Box plots for **b** and **c** present min to max, median, and all points. For **d**, **e**, and **g**, data show mean ± SD and all points. Statistical significance was determined by a two-sided unpaired Student's *t*-test (**b–g**). †$p < 0.0001$, n.s; not significant, between genotypes or groups.

various causes (Supplementary Table 1). Expression of lincRNA transcript 1 was significantly and inversely correlated with that of *NPPB*, a marker of left ventricular dysfunction ($\rho = -0.77$, $p = 0.016$) (Supplementary Fig. 16a), although we observed no correlation between expression of lincRNA transcript 3 and *NPPB* ($\rho = -0.1$, $p = 0.798$) (Supplementary Fig. 16b). Immunohistochemical analysis revealed abundant HINT1 protein levels in heart tissue from cases 1 and 2, accompanied by low levels of lincRNA transcript 1 expression and high *NPPB* expression. Conversely, HINT1 protein levels in heart tissue from cases 3 and 4, which showed high lincRNA transcript 1 level and low *NPPB* expression, were lower than those seen in cases 1 and 2 (Supplementary Fig. 16c).

We then examined lincRNA transcript 1 expression in isoproterenol (Iso)-treated human iPSC-CMs (Supplementary Fig. 16d) and found transcript levels to be significantly decreased relative to those seen in untreated control cells. We then transduced human iPSC-CMs with one of two human lincRNA transcript 1-specific siRNAs (siLincRT1-A and siLincRT1-B) or negative control siRNA (siScramble) and analyzed mitochondrial respiratory capacity using a Cell Mito Stress test (Supplementary Fig. 16e). Basal and maximal respiration levels and ATP production in human lincRNA transcript 1-knockdown iPSC-CMs were significantly decreased relative to values seen in control cells (Supplementary Fig. 16f, g). These results suggest that lincRNA transcript 1 could be a candidate human *Caren* homolog.

## Discussion

Here we used a gene trapping approach and identified *Caren* as a cytoplasmic lncRNA that protects against the development of cardiac hypertrophy and failure under pressure overload. *Caren* transcript levels were relatively decreased in hearts of aged relative to younger mice and in mice subjected to TAC surgery or Ang II infusion relative to untreated controls. Loss-of-function and gain-of-function studies in mice supported the idea that *Caren* protects animals from maladaptive myocardial remodeling during pressure overload. Mechanistic analysis suggested that *Caren* counteracts HF development by suppressing the DDR, protecting against mitochondrial impairment, and enhancing mitochondrial biogenesis. Furthermore, the proteomic analysis identified Hint1 as a cardiac *Caren* target, and *Hint1*^+/−^ mice exhibited anti-HF effects against pressure overload similar to CAG-*Caren* Tg mice.

Recently, it was reported some lncRNA transcripts are translated into functional micropeptides[39–42]. This finding indicates potential misannotation of the non-canonical open reading frame (ORF)-containing genes as ncRNAs. We also observed a potential ORF starting from an ATG within *Caren* that, if translated, would encode a 10aa peptide. Although this ORF is contained within the CAG-*Caren* fragment A in engineered Tg mice analyzed in Supplementary Fig. 13, those mice were not resistant to TAC-induced HF development, strongly suggesting that if expressed this small peptide does not alter cardiac function but rather that *Caren* transcripts themselves counteract HF development.

Previous transcriptome profiling based on deep RNA sequencing showed that cardiac lncRNAs likely function by regulating the expression of nearby (*cis*) rather than distant (*trans*) genes[3,43]. Moreover, nuclear lncRNAs act as important transcriptional regulators by promoting transcriptional interference and/or chromatin remodeling[31,44]. In support of this idea, the most previously characterized cardiac lncRNAs are nuclear[45–49]. *Caren*, however, is predominantly cytoplasmic, and *Caren's* loss does not alter the expression of flanking *Hmgb2* and *Sap30* genes. Our study provides evidence that *Caren* post-transcriptionally suppresses Hint1 expression in the cytoplasm and decreases Hint1 protein abundance. Thus, the cytoplasmic *Caren* lncRNA functions by regulating the translation of distant genes.

We identified Hint1 as a cardiac *Caren* target, although Hint1's cardiac function is largely unknown. However, Hint1 reportedly functions as a tumor suppressor by activating ATM and the subsequent DDR[20]. Interestingly, ATM activation has recently received much attention as a DDR regulator closely linked to HF development[17]. Here, we found that Hint1 is upregulated in the TAC heart. Moreover, ATM activation significantly decreased in hearts of *Hint1*^+/−^ compared to WT controls, and *Hint1*^+/−^ mice resisted HF development. These findings strongly suggest that Hint1 suppression by *Caren* antagonizes HF development through suppressing the ATM-DDR pathway in the heart.

Recent single cardiomyocyte analysis demonstrated that HF development is associated with mitochondrial impairment and the DDR[50]. Our results suggest that *Caren* regulates both mitochondrial impairment and the DDR. For example, our *Caren*^β-geo/β-geo^ mice show increased Hint1 protein levels and reduced mitochondrial respiratory capacity in the heart. Moreover, Hint1 overexpression in cardiomyocytes reduced mitochondrial respiratory capacity. Given that *Caren* prevents Hint1 upregulation during pressure overload, *Caren* may maintain mitochondrial respiratory capacity by inhibiting Hint1-mediated mitochondrial dysfunction, which in turn may antagonize HF development. Furthermore, we consistently observed inhibition of ATM-DDR signaling during pressure overload in both *CAG-Caren* Tg and *Hint1*^+/−^ mice, suggesting that *Caren* blocks Hint1 upregulation brought on by pathological stress and protects against HF development through (1) maintaining mitochondrial respiratory capacity and (2) suppressing ATM-DDR signaling.

Of note, CAG-*Caren* Tg mice and mice injected with AAV6-*Caren* exhibited increased expression of genes functioning in mitochondrial biogenesis and OXPHOS, phenotypes not seen in the *Hint1*^+/−^ the heart. Thus, *Caren* may directly regulate genes/proteins involved in cardiac mitochondrial biogenesis in a Hint1 pathway-independent manner. PGC-1α, NRF1/2, TFAM, and estrogen-related receptor α (ERRα) reportedly play key roles in

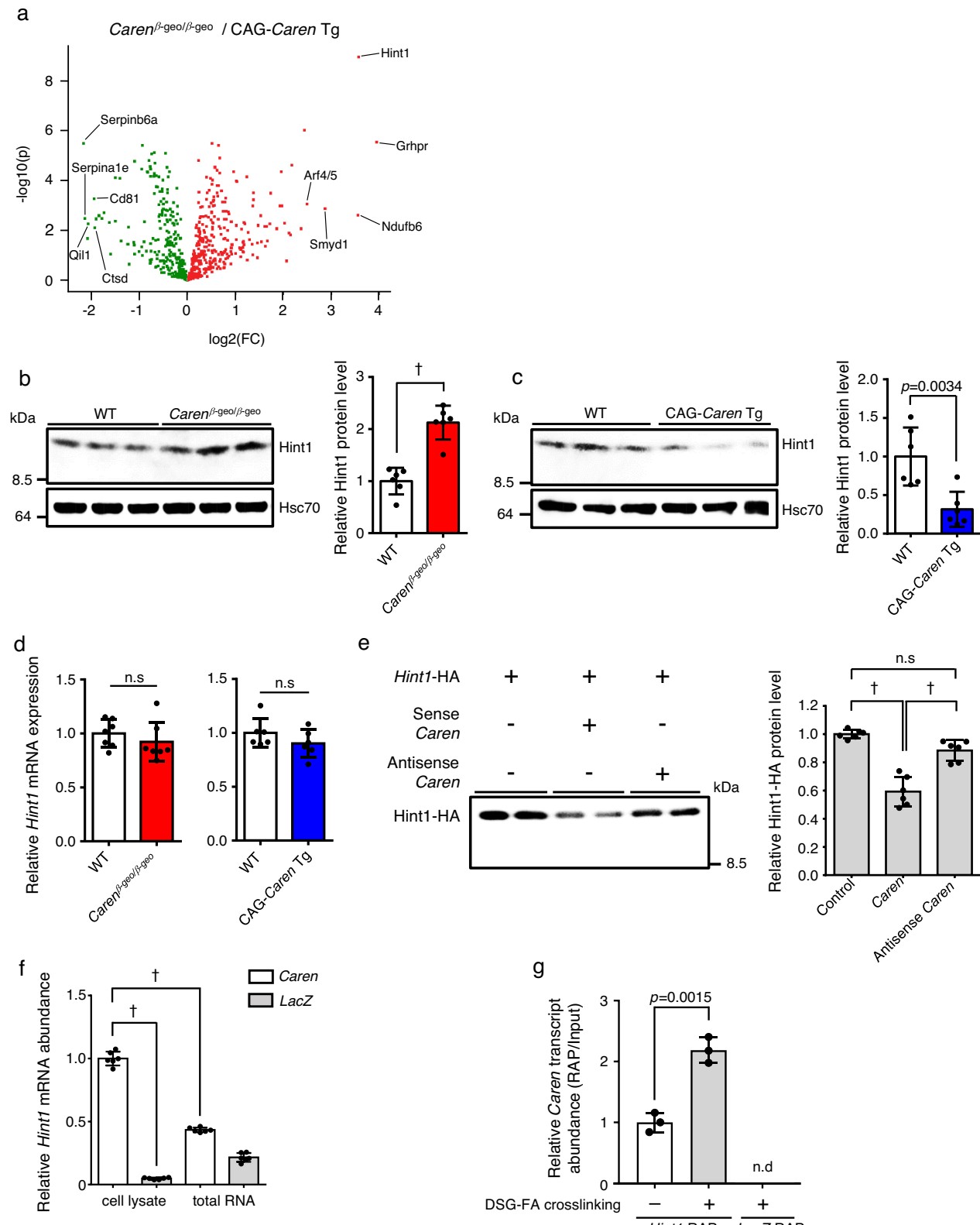

regulating the expression of genes functioning in mitochondrial biogenesis and OXPHOS[51,52]. Interestingly, CAG-*Caren* Tg mice showed increased cardiac expression of *Nrf2* and *Tfam*, suggesting that *Caren* promotes the expression of factors functioning in mitochondrial biogenesis and OXPHOS. Since *Caren* regulates Hint1 translation, it may regulate the translation of transcriptional regulators of *Nrf2* and *Tfam*.

Many reports support the idea that lncRNA function is not solely dependent on primary sequence but rather on higher-order secondary or tertiary structure[53–55]. Indeed, we did not observe anti-HF phenotypes in analyzing any of the five CAG-*Caren* fragments (A, B, C, D, or E) Tg lines described here relative to CAG-*Caren* Tg mice. Moreover, *Hint1* mRNA translation was not altered in the presence of any of those fragments or after

**Fig. 5 *Caren* negatively regulates Hint1 translation. a** Volcano plots showing protein expression differences in 12-week-old CAG-*Caren* Tg vs. *Caren*[β-geo/β-geo] mice (*n* = 7 each). The 5 most significantly altered proteins in *Caren*[β-geo/β-geo] compared to CAG-*Caren* Tg mice are indicated. The *x*-axis indicates fold-change (FC) on a log2 scale, and the *y*-axis indicates *p*-value (*p*) on a −log10 scale. Statistical significance was determined by a two-sided Student's *t*-test without correcting for multiple comparisons. **b**, **c** Representative western blots (left) and quantitation (right) of Hint1 protein levels in heart tissues of 12-week-old *Caren*[β-geo/β-geo] and WT littermate mice (**b**), and of CAG-*Caren* Tg and WT littermate mice (**c**) (*n* = 6 per group). Hsc70 served as the loading control. Protein intensity values of WT mice heart tissue were set to 1. **d** Relative expression of *Hint1* transcripts in heart tissues of 12-week-old *Caren*[β-geo/β-geo] (*n* = 7 per group) and WT littermate mice (left), and of CAG-*Caren* Tg and WT littermate mice (light) (*n* = 6 per group). Values in WT were set to 1. **e** Representative western blot (left) and quantitation (right) of in vitro translated *Hint1* mRNA (*n* = 6 per group). **f** RNA pull-down assay using whole-cell lysates or total RNA extracted from C2C12 cells (*n* = 6 per group). Biotinylated *Caren* transcripts and *LacZ* mRNA served as respective bait RNAs. Levels of *Hint1* mRNA co-precipitated with *Caren* transcripts in cell lysate assays were set to 1. **g** RAP analyses using specific probes for *Hint1* mRNA (*Hint1* RAP) or *LacZ* mRNA (*LacZ* RAP) in P19.CL6 cells (*n* = 3 per group). Values derived from *Hint1* RAP using non-crosslinked cells were set to 1. Data show mean ± SD and all points for **b**–**g** Statistical significance was determined by two-sided unpaired Student's *t*-test (**b**–**d**, **g**) or one-way ANOVA with Sidak's post hoc test (**e**, **f**). †*p* < 0.0001, n.s; not significant, between genotypes or groups.

*Caren* denaturation. These findings support the idea that the *Caren* structure likely mediates its anti-HF effects.

Interestingly, cardiomyocyte size in hearts from sham-operated *Caren*[β-geo/β-geo] mice was slightly but significantly increased compared to that seen in hearts from sham-operated WT mice (Fig. 2e, f), although we observed no difference in expression levels of HF or cardiac fibrosis markers between genotypes (Fig. 2g). By contrast, sham-operated CAG-*Caren* Tg mice showed decreased cardiomyocyte size compared with sham-operated WT mice (Fig. 3e, f). Moreover, consistent with effects seen in CAG-*Caren* Tg mice, cardiomyocyte size in hearts from sham-operated *Hint1*[+/−] mice was slightly but significantly decreased compared to hearts from sham-operated WT mice (Fig. 7e, f). These findings suggest that even in some non-pathological states, *Caren* regulates cardiomyocyte homeostasis by maintaining Hint1 expression within a normal physiological range.

HF is a leading cause of death in developed countries. The number of HF patients continues to grow in parallel with the size of the elderly population worldwide[56–58]. Despite new therapies and considerably improved survival in recent years, mortality remains high, with approximately 50% of HF patients dying within 5 years of diagnosis. Thus, therapeutic approaches are urgently needed. It is critical to define molecular mechanisms underlying HF pathogenesis in order to develop those therapies. Importantly, our findings suggest that lincRNA transcript 1 could be a candidate human *Caren* homolog, encouraging future analysis of potential cardioprotective mechanisms by this and other human lincRNAs.

In conclusion, ours is the first study showing that the cytoplasmic lncRNA *Caren* maintains cardiac function during pressure overload potentially by suppressing the ATM-DDR pathway and maintaining mitochondrial function by enhancing mitochondrial biogenesis and suppressing Hint1 expression. Our findings open the door for investigation of how cytoplasmic lncRNAs function in physiological or pathological contexts.

## Methods

**Animal studies**. All experimental procedures were approved by the Kumamoto University Ethics Review Committee for Animal Experimentation (approval No. A27-063, A29-072, and A2019-063). All animals were fed a normal diet (ND; CE-2, CLEA, Tokyo, Japan), bred in a mouse house with automatically controlled lighting (12 h on, 12 h off), and maintained at a stable temperature of 22 ± 2 ℃ and relative humidity of 40–80%.

**Transverse aortic constriction (TAC) surgery**. Male mice approximately 12 weeks old (bodyweight of 28–32 g) were subjected to TAC surgery, as described[32,33]. Briefly, the transverse aorta was constricted with an 8-0 monofilament suture in parallel with a 27-gauge blunt needle, which was removed after the constriction. Control sham-operated mice underwent similar surgery without aortic constriction. TAC surgery was performed on randomly selected mice by a

surgeon unaware of their genotypes. After surgery, transthoracic echocardiography was performed using a high-frequency ultrasound system dedicated to small animal imaging (VisualSonics Vevo 2100, FujiFilm VisualSonics, Toronto, Canada) using an MS 400 linear array transducer (18–38 MHz). M-mode recording was performed at the midventricular level. All images were analyzed using dedicated software (Vevo 2100 version 1.4). LV wall thickness and internal cavity diameters at diastole (LVD:d) and systole (LVD:s) were measured. Percent LV fractional shortening (%FS), left ventricular end-diastolic volume (LVEDV), left ventricular end-systolic volume (LVESV), cardiac output (CO), percent ejection fraction (% EF), stroke volume (SV), and heart rate (HR) were calculated from M-mode measurements automatically. For echocardiography, inhalation anesthesia was used. Specifically, isoflurane was mixed with air, diluted to 1%, and inhaled using Vevo Compact Dual Anesthesia System: Tabletop Version (FujiFilm VisualSonics). After the righting reflex disappeared, the animal was fixed in a supine position on a heating pad (Vevo® Integrated Rail System, FujiFilm VisualSonics) to maintain normothermia and to allow placement of electrocardiographic limb electrodes. Mice were allowed to breathe spontaneously, and anesthesia was maintained with 0.5–2% isoflurane to keep the heart rate between 550 ± 50 bpm. All procedures were performed under double-blind conditions with regard to genotype or treatment.

**Angiotensin II treatment**. Angiotensin II (Ang II, Peptide Institute, Osaka, Japan) was dissolved in 150 mM NaCl and 1 mM acetic acid. Dorsal subcutaneous tissues of mice (male, 10-week-old) were implanted with a mini-osmotic pump (Model 2004, Alzet Corp., Palo Alto, CA, USA) to infuse Ang II (3 mg/kg per day) continuously for 2 weeks. Vehicle-treated groups underwent the same procedure using 150 mM NaCl and 1 mM acetic acid (vehicle)-filled pumps.

**ATM inhibitor treatment**. Two days after TAC, *Caren*[β-geo/β-geo] mice were intraperitoneally-administered the ATM inhibitor KU-60019 (Selleck Chemicals, Houston, TX, USA) at 10 mg/kg every 3 days for 6 weeks. Cardiac function was evaluated by echocardiography before TAC surgery and at 2, 4, and 6 weeks afterward. Control groups underwent the same procedure using phosphate-buffered saline (PBS).

**Recombinant adeno-associated virus (AAV) treatment**. To analyze TAC animals, 10-week-old male C57BL/6N mice (CLEA) were subjected to TAC surgery and 1 week later anesthetized with 2% isoflurane and intravenously injected with recombinant AAV6 vectors at $1 \times 10^{11}$ vg. Cardiac function was examined using echocardiography before injection and at 2, 3, 4, 6, and 8 weeks afterward. In some experiments, 10-week-old male C57BL/6N or *Caren*[β-geo/β-geo] mice were subjected to TAC surgery and 1 week later intravenously injected with recombinant AAV6 vectors at $1 \times 10^{11}$ vg. Mouse survival was monitored at various time points thereafter.

**Generation of gene trap mice**. Isolation of the T167 gene trap line was mentioned in the results section (Fig.1a–c and Supplementary Fig. 1a, b). Chimeric mice were produced by aggregating ES cells with ICR eight-cell embryos. Chimeric male mice were backcrossed to C57BL/6N females to obtain F1 heterozygotes. Genotyping was performed by PCR of tail DNA using the following primers: T167-S (5'-GGAG GGAAATCTATAGGATCC-3') and T167-AS (5'-CCAAGCTACTGGAATACTC TG-3') to detect the wild-type allele; and Z-S1 (5'-GCGTTACCCAACTTAATC-3') and Z-AS1 (5'-TGTGAGCGAGTAACAACC-3') to detect the mutant allele. In this study, we used F9 generation mice. *Caren*[β-geo/β-geo] mice were maintained using heterozygous breeders.

**Generation of *CAG-Caren* and *CAG-Caren*-(fragment A–E) transgenic mice**. *Caren*-overexpressing mouse lines were produced through *Cre*-mutant *lox*-

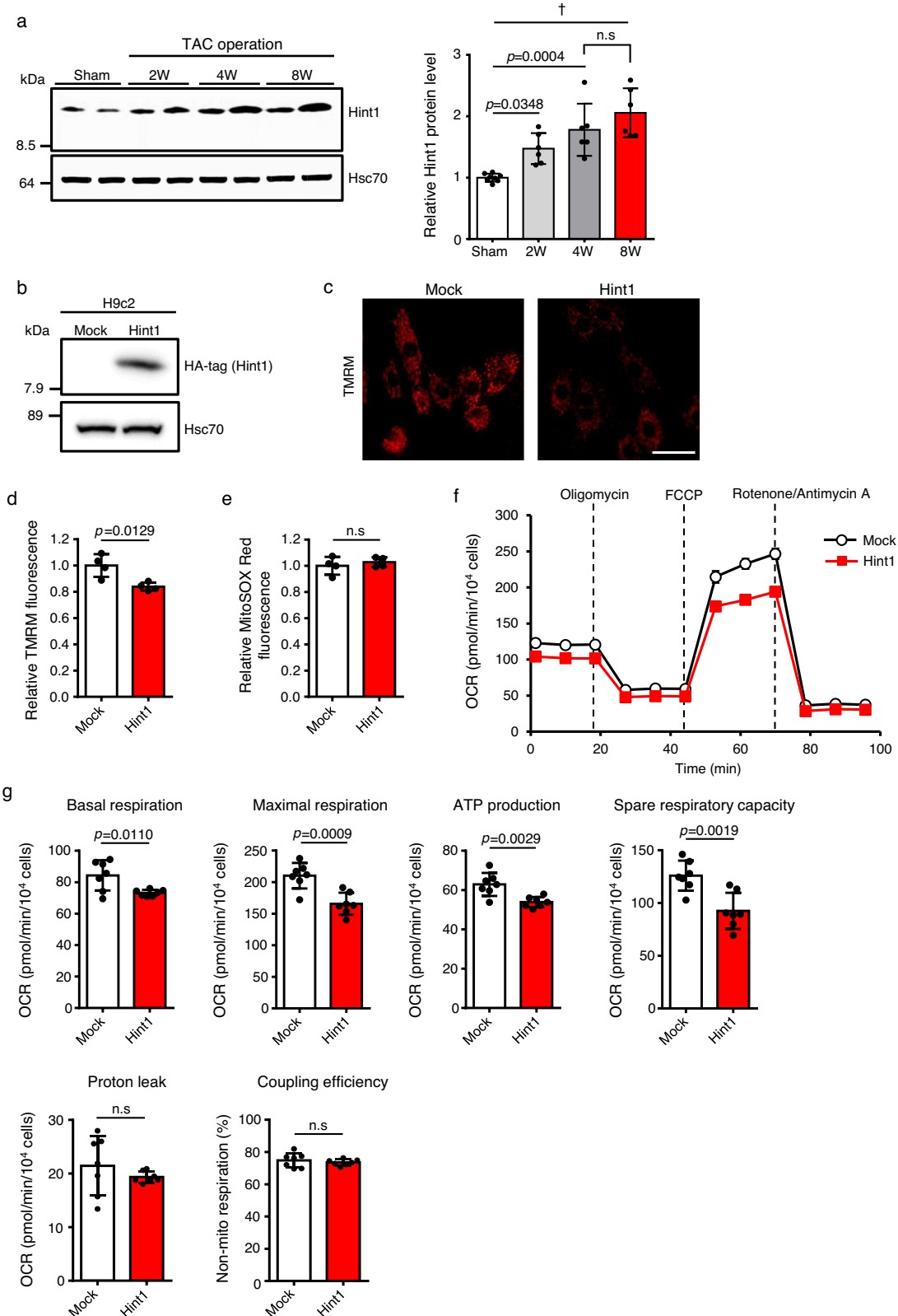

mediated cassette exchange as described[12]. T167 ES cells were used to replace the *β-geo* gene with the CAG-*Caren* cassette (Supplementary Fig. 4a) or with a CAG-*Caren*-(fragment A-E) cassette (Supplementary Fig. 13a, b). ES cells were co-electroporated with the replacement vector and pCAGGS-*Cre* vector[59]. Puromycin-resistant colonies were analyzed by PCR to select ES clones showing successful cassette exchange. Positive clones were aggregated with ICR eight-cell embryos. Chimeric male mice were backcrossed to C57BL/6 N females to obtain F1 heterozygotes. To identify Tg offspring, genomic PCR was performed with forward

(5'-CATTCCTTCCGCTAGCTTTTATTG-3') and reverse (5'-AAGTCCCATAAG GTCATGTACTGG-3') primers. To maintain an isogenic strain, mice were propagated as heterozygotes by breeding with WT C57BL/6 N mice.

**Generation of *αMHC-Caren* transgenic mice.** Mouse *Caren* cDNA was cloned into the murine *αMHC* promoter expression vector[32,60] (Supplementary Fig. 5a). To identify Tg offspring, genomic PCR was performed with forward (5'-CAG-GACTTCACATAGAAGCC-3') and reverse (5'-CCAGGCAGTCATGATGCAGG-

**Fig. 6 Increased Hint1 levels in cardiomyocytes reduces mitochondrial respiratory capacity. a** Representative western blots (left) and quantitation (right) of Hint1 protein levels in heart tissues of 12-week-old sham-operated WT mice and at 2, 4, and 8 weeks (W) after TAC in WT mice (sham: $n = 8$, other groups: $n = 6$ each). Hsc70 served as the loading control. Protein intensity values of sham-operated heart tissue were set to 1. **b** Representative immunoblotting of HA-tagged Hint1 protein in H9c2-mock and H9c2-Hint1 cells. Hsc70 served as the loading control. This experiment was repeated independently three times with similar results**. c** Representative TMRM fluorescence images of H9c2-mock and H9c2-Hint1 cells (scale bar, 50 μm). **d, e** Quantification of TMRM (**d**) and MitoSOX Red (**e**) fluorescence intensity in H9c2-mock and H9c2-Hint1 cells ($n = 4$ per group). Values derived from H9c2-mock cells were set at 1. **f** Mitochondrial oxygen consumption rate (OCR) in H9c2-mock and H9c2-Hint1 cells ($n = 7$ biological replicates per group). **g** Quantification of basal respiration, maximal respiration, ATP production, spare respiratory capacity, proton leak, non-mitochondrial respiration, and coupling efficiency in H9c2-mock and H9c2-Hint1 cells ($n = 7$ biological replicates per group). Data show mean ± SD and all points for **a, d, e, g**. For **f**, data show mean ± SD. Statistical significance was determined by two-sided unpaired Student's $t$-test (**d, e, g**) or one-way ANOVA with Sidak's post hoc test (**a**). [†]$p < 0.0001$, n.s; not significant, between groups.

3') primers. To maintain an isogenic strain, mice were propagated as heterozygotes by breeding with WT C57BL/6N mice.

**Generation of *Hint1* KO mice**. Mutagenized sperm from mice on a C57BL/6N background (EM:05220) was purchased from the European Mouse Mutant Archive (EMMA, http://www.infrafrontier.eu/) and used for in vitro fertilization with C57BL/6N eggs. To generate *Hint1* KO mice through Cre/*lox*-mediated recombination, pronuclear embryos were electroporated with *Cre* mRNA (20 ng/μl) using the Super Electroporator NEPA 21 (NEPA GENE, Chiba, Japan; Supplementary Fig. 10a). Genotyping was performed by PCR of tail DNA using the following primers: Hint1-S (5'-GACATTTCCCCTCAAGCACCAACACACTTTCTGG-3') and Hint1-AS (5'-GCACCTTCATTCACCACCATCCG-3') to detect the wild-type allele; and Z-S1 (5'-GCGTTACCCAACTTAATC-3') and Z-AS1 (5'-TGTGAGCG AGTAACAACC-3') to detect the mutant allele. To maintain the strain, mice were propagated as heterozygotes by breeding with C57BL/6N mice.

**β-galactosidase assay**. Whole-mount X-gal staining was performed as described by Allen et al.[61]. Samples were fixed 30 min at room temperature 1% formaldehyde/0.2% glutaraldehyde/0.02% NP-40 in PBS. Samples were cut in the sagittal plane with a blade after 15 min of fixation. Samples were washed once in 1% TritonX-100/PBS and twice in PBS and incubated overnight at room temperature in staining solution (5 mM potassium ferricyanide, 5 mM potassium ferrocyanide, 0.5 mM MgCl2, 0.5% X-gal in PBS). Samples were rinsed three times in PBS, post-fixed in 3.7% formaldehyde/PBS, and made transparent using benzylalcohol/benzylbenzoate (1:2) after dehydration with a series of ethanol steps (25, 50, 70, 100, and 100%, 1 h each). In some experiments, adult heart tissues were fixed in 3.7% paraformaldehyde/PBS for 1 h and then washed three times with PBS. Samples were incubated overnight at 30 °C in staining solution, rinsed three times in PBS, post-fixed in 3.7% formaldehyde/PBS for 24 h, and embedded in paraffin. Blocks were cut into 4 μm thick sections, air-dried, and deparaffinized. Sections were stained with nuclear fast red (Wako, Osaka, Japan). Slides were mounted and examined using a BZ-X710 microscope (Keyence, Osaka, Japan).

**RT-PCR analysis**. Total RNA was extracted using an RNeasy Mini Kit (Qiagen, Valencia, CA, USA). DNase-treated RNA was reverse transcribed using a Prime-Script RT reagent Kit (Takara Bio Inc, Shiga, Japan). Heart tissue was homogenized using a multi-beads shocker (Yasui Kikai, Osaka, Japan). Real-time quantitative RT-PCR was performed using TB Green Premix Ex Taq II, Premix Ex Taq (Probe qPCR) (Takara Bio Inc), and a Thermal Cycler Dice Real-Time System (Takara Bio). Relative transcript abundance was normalized to that of *18 S* rRNA levels in mouse and human samples. Primer sets used for RT-PCR are listed in Supplementary Table 2.

**Quantification of mitochondrial DNA content**. To examine mitochondrial DNA (mtDNA) content, mtDNA/nuclear DNA (nDNA) ratios were determined. In brief, DNA was extracted from the heart tissues and subjected to real-time quantitative PCR analysis using specific primer sets for *CytB* (mtDNA) and *β-actin* (nDNA) (Supplementary Table 3). Real-time quantitative PCR was performed using TB Green Premix Ex Taq II (Takara Bio Inc) and a Thermal Cycler Dice Real-Time System (Takara Bio).

**Isolation of cardiomyocytes and non-cardiomyocytes**. To isolate cardiomyocytes and non-cardiomyocytes, ventricles were harvested from 10-week-old male Tg mice overexpressing enhanced green fluorescent protein (EGFP) driven by the murine α*MHC* (*Myh6*) promoter (α*MHC-EGFP*)[32], and tissue was minced and digested with 0.075% collagenase, 0.12% trypsin and 0.02% DNase at 37 °C for 40 min. Cells were collected, resuspended, and then passed through a 100-μm mesh filter into 50-ml centrifuge tubes. Cells were stained with 7-AAD and finally resuspended in 0.5 ml FACS buffer (phosphate-buffered saline (PBS)/0.1% BSA) and GFP-positive (cardiomyocyte) and GFP-negative (non-cardiomyocyte) viable cells were isolated using a cell sorter FACSAria II (Becton Dickinson, San Jose, CA, USA).

**Analysis of nuclear and cytosolic transcripts**. Heart tissue was homogenized using a multi-beads shocker (Yasui Kikai). RNA in homogenized heart tissue lysates was separated and purified using a Cytoplasmic and Nuclear RNA Purification Kit (Norgen Biotek, Thorold, Canada). DNase-treated cytoplasmic or nuclear RNA was subjected to quantitative RT-PCR analysis.

**Western blotting**. Mouse heart tissue was homogenized in lysis buffer (10 mM Tris–HCl, 1% Triton X-100, 50 mM NaCl, 30 mM sodium pyrophosphate, 50 mM NaF, 5 mM EDTA, 0.1 mM Na3VO4, plus a protease inhibitor cocktail (Nacalai Tesque, Kyoto, Japan), pH 7.5) using a multi-beads shocker (Yasui Kikai). Proteins (20 mg) were separated by SDS–PAGE and transferred to PVDF membranes. Membranes were incubated with anti-phosphorylated ATM (Ser1981) (10H11.E12, #4526, Cell Signaling Technology, Danvers, MA, USA), anti-ATM (D2E2, #2873, Cell Signaling Technology), anti-HINT1 (ab124912, Abcam, Cambridge, MA, USA), anti-HMGB2 (ab67282, Abcam), anti-SAP30 (ab125187, Abcam), anti-HA-tag (C29F4, #3724, Cell Signaling Technology), anti-TFAM (ab131607, Abcam), or anti-Total OXPHOS Rodent Antibody Cocktail (ab110413, Abcam) containing the antibodies against NDUFB8 (Complex I, ab110242), SDHB (Complex II, ab14714), UQCRC2 (Complex III, ab14745), MTOCI (Complex IV, ab14705), and ATP5A (Complex V, ab14748). Antibodies were diluted 1:1000, and samples were incubated at 4 °C overnight. After TBST washing, membranes were incubated with 1:2000 diluted horseradish peroxidase (HRP)-conjugated donkey anti-rabbit IgG or sheep anti-mouse IgG (GE Healthcare Life Science, Piscataway, NJ, USA) antibodies at room temperature for 60 min. Internal controls were incubated with 1:2000 diluted anti-Hsc70 (sc-7298, Santa Cruz Biotechnology, Santa Cruz, CA, USA) and 1:2000 diluted HRP-conjugated sheep anti-mouse IgG (GE Healthcare Life Science) antibodies, which were used as primary and secondary antibodies, respectively. Blots were incubated with ECL Western Blotting Detection Reagent (GE Healthcare Life Science), visualized using a Luminescent Image Analyzer LAS-4000 system (Fujifilm, Tokyo, Japan), and quantified with Multi Gauge software version 3.1 (Fujifilm).

**Transmission electron microscopy**. Mouse hearts were fixed in phosphate buffer containing 2% paraformaldehyde and 2.5% glutaraldehyde (pH 7.4) at 4 °C for 4 h. After washing in phosphate buffer at 4 °C, specimens were postfixed in 1% osmium tetraoxide at 4 °C for 2 h. Specimens were then washed repeatedly in distilled water, stained with 1% uranyl acetate for 30 min, dehydrated through a graded ethanol series and propylene oxide, and embedded in Glicidether (Selva Feinbiochemica, Heidelberg, Germany). Ultrathin sections were cut and mounted onto nickel grids, stained with 1% uranyl acetate for 10 min and with Reynolds lead citrate for 5 min, and then observed using an HT7700 electron microscope (Hitachi High-Tech Corporation, Tokyo, Japan)[62].

**AAV production**. To construct AAV plasmids encoding *Caren* or *GFP*, respective cDNAs were cloned into the pAAV-CMV vector (Takara Bio Inc). Recombinant AAV6 vectors were produced with an AAVpro Helper Free System (Takara Bio Inc) and purified using an AAVpro Purification Kit (Takara Bio Inc), according to the manufacturer's instructions. Genome copy number was determined using an AAVpro Titration Kit (for Real-time PCR) Ver. 2 (Takara Bio Inc).

**Proteomic analysis**. For sample preparation, protein extraction from murine heart tissue from 12-week-old *Caren*[β-geo/β-geo], CAG-*Caren* Tg, or corresponding littermate mice was carried out by lysing the tissue in lysis buffer (0.1 M Tris-HCl, 4% SDS, 0.1 M DTT, pH 7.6) and then sonicating the samples with a tip sonicator for 3 cycles of 10 s and incubating at 95 °C for 10 min. The lysate was then centrifuged at 16,000×*g* for 15 min to remove any insoluble material. Lysate (200 μl) from each sample was loaded into a Vivacon 500 filter units (Sartorius), concentrated (10-fold) at 13,000 × *g*, and then washed three times with 100 μl of UA buffer (8 M Urea, 0.1 M Tris-HCl, pH 8.5). The concentrate was mixed with 100 μl of 50 mM iodoacetamide in UA buffer and incubated in darkness at room temperature for 20 min followed by centrifugation for 15 min (13,000 × *g*). The concentrate was then washed twice with 100 μl of UA buffer followed by two washes with 100 μl of

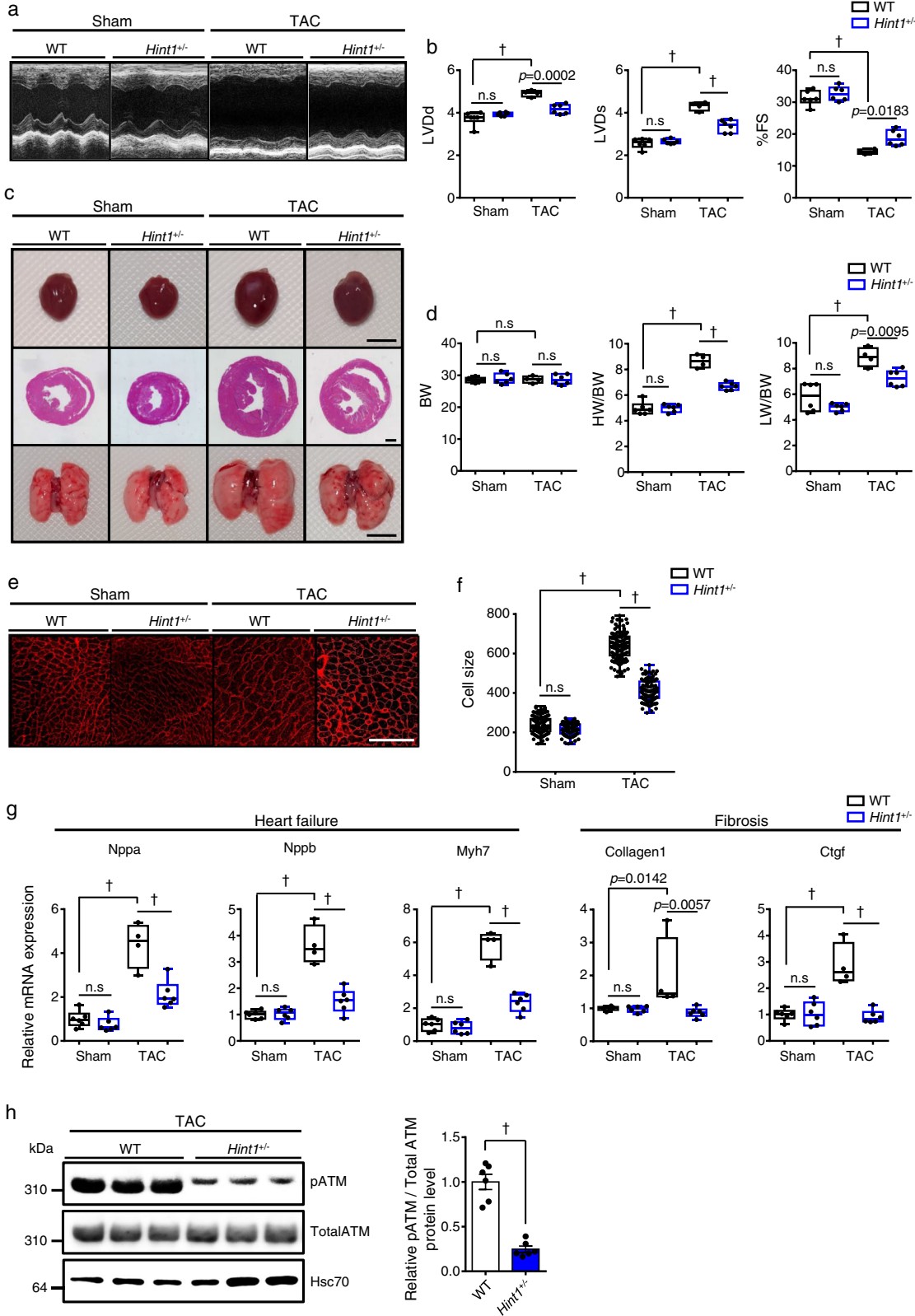

50 mM ammonium bicarbonate. Approximately 400 µg of protein from each sample was mixed with trypsin (1:40) in the filter and incubated overnight at 37 °C. The peptides were then eluted with 50 mM ammonium bicarbonate and acidified with 1% formic acid.

For LC-MS/MS analysis, proteomic analysis was carried out using a liquid chromatography/tandem mass spectrometry (LC-MS/MS) system. Samples were diluted in 50 µl of 0.1% formic acid, and the peptides were separated on a 15 cm long, 75 µm ID reversed-phase PicoFrit column packed with 3 µm Reprosil-Pur

C-18 resin at 400 nl/min, using gradient formed by the mixture of A (0.1% FA, 5% DMSO) and B (0.1% FA, 5% DMSO, and 95% acetonitrile) and duration of 255-min (from 0–12% B in 20 min, 30–45% B in 210 min and 28–80% B in 25 min) gradient. The MS/MS analysis was carried out on an LTQ-Orbitrap Velos Pro instrument (Thermo Fisher Scientific). The MS1 scans were acquired in the Orbitrap analyzer with 60,000 resolution at $m/z$ of 400. The AGC targets were set at $8 \times 10^6$ for MS1 scans $3 \times 10^4$ for MS2 scans. The 20 most intense ions in each full MS1 scan were fragmented using CID with a normalized collision energy of 35.

**Fig. 7 *Hint1*-deficient mice are resistant to HF development. a** Representative M-mode echocardiography recordings from *Hint1*[+/−] and WT littermate mice, 6 weeks after TAC or sham surgery. **b** Left ventricular end-diastolic diameter (LVDd) (mm) (left), left ventricular end-systolic diameter (LVDs) (mm) (middle), and percent fractional shortening (%FS) (right) in indicated mice (WT TAC: $n = 4$, other groups: $n = 6$ each). **c** Gross appearance of whole heart (top row; scale bar, 5 mm), hematoxylin-eosin (HE)-stained sections of the mid-portion of the heart (middle row; scale bar, 1 mm), and gross appearance of the whole lung (bottom row; scale bar, 5 mm). **d** Body weight (BW) (g) (left), HW/BW ratio (mg/g) (middle), and lung weight per body weight ratio (LW/BW) (mg/g) (right) in indicated mice (WT TAC: $n = 4$, other groups: $n = 6$ each). **e** Shown are representative left ventricle sections stained with wheat germ agglutinin (WGA) as an indicator of cardiomyocyte size (scale bar, 100 μm). **f** Distribution of myocardial cell size (μm$^2$) in indicated mice (WT sham: $n = 117$ cells, *Hint1*[+/−] sham: $n = 113$ cells, WT TAC: $n = 112$ cells, *Hint1*[+/−] TAC: $n = 116$ cells). **g** Relative expression of genes associated with heart failure and fibrosis in hearts of indicated mice (WT TAC: $n = 4$, other groups: $n = 6$ each). Levels seen in sham-operated WT mice were set to 1. **h**, Representative western blot (left) and pATM/total ATM ratio (right) in heart tissues of WT or *Hint1*[+/−] littermate mice, 6 weeks after TAC operation ($n = 6$ per group). Hsc70 served as a loading control. Values derived from TAC-operated WT mouse heart tissue were set to 1. Box plots for **b**, **d**, **f**, and **g** present min to max, median, and all points. For **h**, data show mean ± SD and all points. Statistical significance was determined by one-way ANOVA with Sidak's post hoc test (**b**, **d**, **f**, and **g**) or two-sided unpaired Student's *t*-test (**h**). †$p < 0.0001$, n.s; not significant, between genotypes or groups.

Dynamic exclusion was enabled with a repeat count of 1 and an exclusion duration of 60 s at 15 ppm mass tolerance. Each biological replicate was run in duplicate.

For data analysis, raw MS files were searched against the Uniprot mouse database with MaxQuant version 1.6.0.16 using the Andromeda search engine. The search included variable modifications for oxidation of methionine, acetylation of protein n-termini, and fixed modification of carbamidomethylation of cysteine. The mass tolerance for MS1 and MS2 searches was 20 ppm and 0.5 Da, respectively. A false discovery rate of 1% was used for both proteins and peptides and only peptides with more than 6 amino acid residues were considered for identification and quantitation. Label-free quantitation was carried out in MaxQuant using the MaxLFQ algorithm. The significance of changes in protein abundance was determined by a Student's two-sided *t*-test with FDR correction of protein LFQ intensity[63,64]. Proteins identified by at least one unique peptide and two or more total peptides, with a *p*-value less than 0.01, were considered significant. Enrichment of biological processes, cellular components, and KEGG pathways was assessed using the STRING database[35], while other downstream data analysis was carried out using Perseus (1.6.0.7)[65].

**Cell culture**. The rat cardiomyocyte cell line H9c2, the mouse myoblast cell line C2C12, and Lenti-X 293 cells (Takara Bio Inc) were maintained in DMEM supplemented with 10% fetal calf serum (FCS), 100 U/ml penicillin, and 100 μg/ml streptomycin under 5% CO$_2$ and 95% air. The mouse embryonal carcinoma cell line P19.CL6 was maintained in MEM-α supplemented with 10% FCS.

**Establishment of a Hint1-overexpressing cell line**. Mouse *Hint1* cDNA fused with an HA-tag (mHint1-HA) was cloned into the pLVSIN-CMV Neo vector (Takara Bio Inc). To produce lentiviral vectors, the lentiviral plasmid encoding mHint1-HA or the pLVSIN-CMV Neo vector was cotransfected with Lentiviral High Titer Packaging Mix (Takara Bio Inc) into Lenti-X 293 cells (Takara Bio Inc) using the TransIT-Lenti Transfection Reagent (Takara Bio Inc), according to the manufacturer's instructions. H9c2 cells were transduced with lentiviral vectors and stable lines were selected with G418 (600 μg/ml).

**Measurement of mitochondrial membrane potential and ROS levels**. To determine mitochondrial membrane potential and mitochondrial ROS levels, cells were treated with 200 nM TMRM (Thermo Fisher Scientific Inc) and 5 μM MitoSOX Red Superoxide Indicator (Thermo Fisher Scientific Inc), respectively, at 37 °C for 30 min. Cells were harvested and analyzed by BD FACSVerse (BD biosciences, San Jose, CA, USA). Data analysis was performed using Flowjo software (BD biosciences). In some experiments, cells seeded in poly-L-lysine-coated 35 mm glass-bottom dishes were treated with TMRM as described above, and images were obtained using a BZ-X710 microscope (Keyence).

**Analyses of lincRNAs in human induced pluripotent stem cell-derived cardiomyocytes (iPSC-CMs)**. iPSC-CMs were purchased from Takara Bio Inc and cultured according to the manufacturer's instructions. Total RNA was extracted using an RNeasy Micro Kit (Qiagen). DNase-treated RNA was reverse transcribed using a PrimeScript RT reagent Kit (Takara Bio Inc). To assess lincRNA transcript expression, semi-nested PCR analysis was performed using KOD FX (TOYOBO LIFE SCIENCE, Osaka, Japan). In brief, cDNA was amplified with 1st PCR primer pairs specific for each lincRNA transcript (Supplementary Table 4), and products were then purified using a QIAquick PCR Purification Kit (QIAGEN) and subjected to a 2nd PCR reaction with 2nd PCR primer pairs specific for each lincRNA transcript (Supplementary Table 4). Second PCR products were analyzed by electrophoresis.

In some experiments, human iPSC-CMs were incubated 12 h with 100 nM isoproterenol and then subjected to real-time quantitative RT-PCR using primer pairs specific for lincRNA transcript 1 (Supplementary Table 2).

**Extracellular flux analysis**. Cellular oxygen consumption rate (OCR) was measured using an XFe24 extracellular flux analyzer (Seahorse Bioscience, North Billerica, MA, USA). H9c2 cells and human iPSC-CMs were seeded at $2.5 \times 10^4$ and $2-3 \times 10^4$ cells per well, respectively, in 24-well Seahorse assay plates (Seahorse Bioscience). One hour prior to the assay, the medium was changed to Seahorse XF DMEM medium (Seahorse Bioscience) supplemented with 10 mM glucose, 2 mM glutamine, and 1 mM sodium pyruvate. To assay mitochondrial respiratory function, the following compounds (final concentrations) were sequentially injected into each well: oligomycin (1 μM), FCCP (4 μM for H9c2 cells and 2 μM for iPSC-CMs), and Rotenone/Antimycin A (0.5 μM each). OCR was measured under basal conditions and after each injection. Data were normalized to cell number.

**Knockdown of lincRNA transcripts in human iPSC-CMs**. Human iPSC-CMs were transfected with lincRNA transcript 1-specific siRNAs: UUCUGGAAC UUAAGCUGAUUCUUUG (siLincRT1-A) and UGGAGAAGGGCGGAGUCA UAUCAUU (siLincRT1-B) (Thermo Fisher Scientific Inc) or with a negative control siRNA (siScramble) (Mission siRNA Universal Negative Control, Sigma-Aldrich, St Louis, MO, USA) using Lipofectamine RNAiMAX (Thermo Fisher Scientific Inc), according to the manufacturer's instructions. Forty-eight hours later, cells were subjected to real-time quantitative RT-PCR analysis or extracellular flux analysis.

**in vitro translation assay**. An in vitro translation assay was performed using a Human Cell-Free Protein Expression System (Takara Bio Inc), according to the manufacturer's instructions. In brief, cDNA encoding C-terminally HA-tagged murine *Hint1* was cloned into the pUC-T7-IRES vector (Takara Bio Inc), and *Caren*, antisense *Caren*, and *Caren* fragment A–E cDNAs were cloned into pSP72 (Promega, Madison, WI, USA). The plasmid encoding *Hint1* and either the pSP72 vector or plasmids encoding *Caren*, antisense *Caren* or one of *Caren* fragments A–E were incubated with T7 RNA polymerase in human cell lysates at 37 °C for 2 h. Samples were then subjected to western blotting analysis. In some experiments, in vitro-transcribed *Caren* was incubated at 94 °C for 2 min and snap-cooled on ice for denaturation[66]. Plasmids encoding *Hint1* plus either native or denatured *Caren* were incubated with T7 RNA polymerase in human cell lysates at 37 °C for 2 h and then subjected to western blotting analysis.

**RNA pull-down assay**. *Caren* and *LacZ* (namely a 3,066-base pair fragment from β-geo) were cloned into a pSP72 plasmid vector (Promega) and transcribed in vitro at 37 °C for 2 h using T7 RNA polymerase (Roche, Indianapolis, IN, USA). Both transcripts were biotinylated using a 5' End-Tag Nucleic Acid Labeling system (Vector Laboratories, MB-9001, Burlingame, CA, USA) and a biotin-maleimide reagent (Vector Laboratories, SP-1501). Each biotinylated transcript (500 ng) was incubated with C2C12 whole-cell lysates (50 μg per sample) or total RNA extracted from C2C12 cells (1 μg per sample) using an RNAeasy mini kit (QIAGEN) in lysis buffer (10 mM Tris–HCl, 1% Triton X-100, 50 mM NaCl, 30 mM sodium pyrophosphate, 50 mM NaF, 5 mM EDTA, 0.1 mM Na$_3$VO$_4$, a protease inhibitor cocktail (Nacalai Tesque), and 200 U/ml RNase inhibitor, pH 7.5) at 32 °C for 1 h, incubated with Dynabeads M-270 Streptavidin (Thermo Fisher Scientific Inc) at 4 °C for 1 h with rotation, and then washed 5 times for 5 min each with 1× SSC/0.5% SDS with rotation at room temperature. Biotinylated transcript-bound RNA was extracted using an RNAeasy mini kit (QIAGEN) and *Hint1* mRNA levels were detected using real-time quantitative RT-PCR.

**RNA antisense purification (RAP) analysis**. To assess cytoplasmic *Caren* transcript–*Hint1* mRNA interactions, we modified the RAP protocol described by Engreitz et al[67]. In brief, P19.CL6 cells were crosslinked with 2 mM disuccinimidyl glutarate for 20 min at room temperature followed by 2% formaldehyde for 10 min at room temperature. Crosslinked or non-crosslinked cells were treated with lysis buffer (20 mM HEPES-KOH, 50 mM KCl, 1.5 mM MnCl$_2$, 1% NP-40, 0.1% N-lauroylsarcosine, 0.4% sodium deoxycholate, 1 mM TCEP, 0.5 mM PMSF, and 80

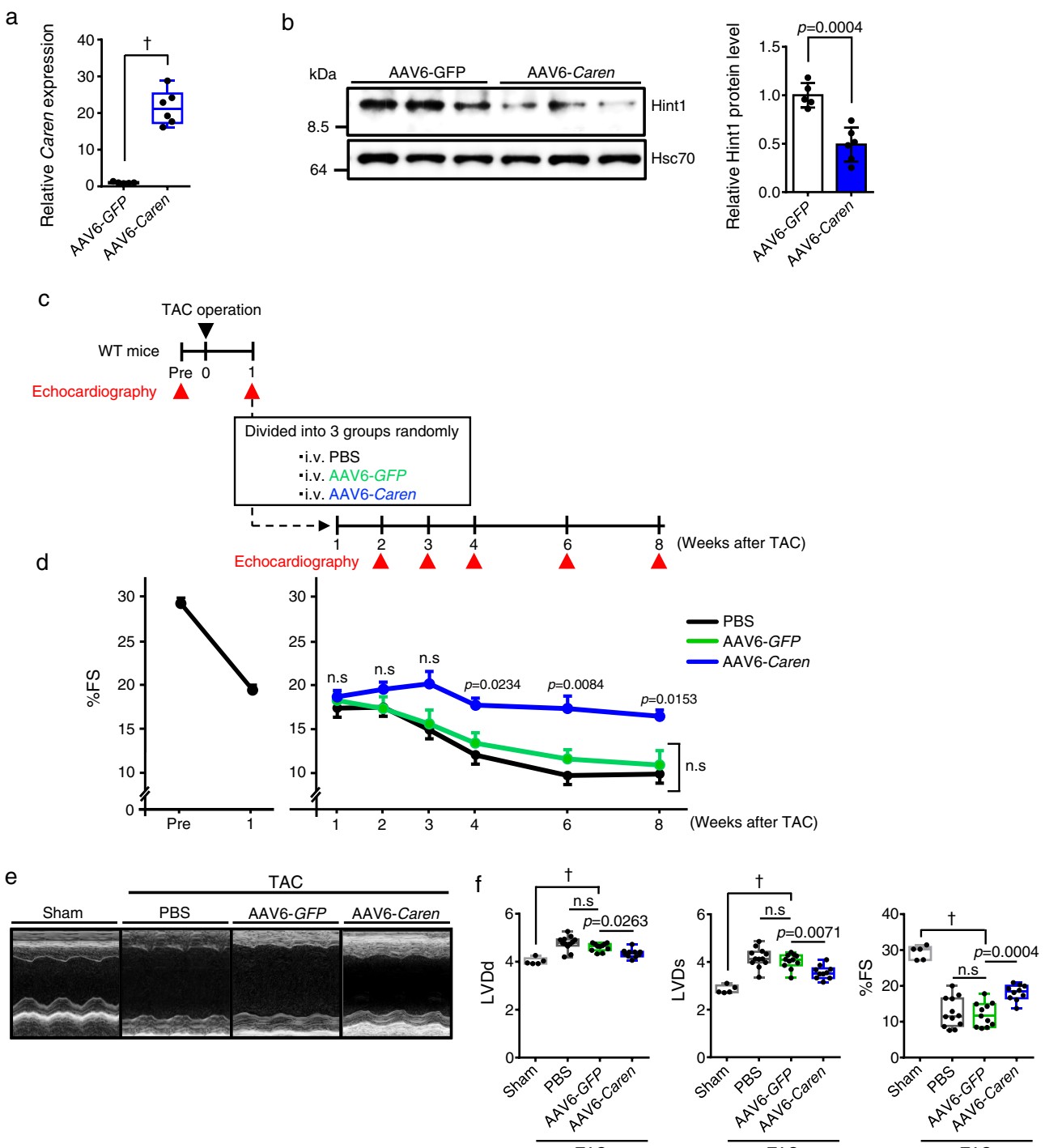

**Fig. 8 Intravenous injection of AAV6-*Caren* slows the progression of heart failure in mice subjected to pressure overload. a** Quantification of *Caren* expression in whole heart tissues of WT mice injected with $1 \times 10^{11}$ vg per mouse of AAV6-*GFP* or AAV6-*Caren*, 1 week after injection (AAV6-*GFP*: $n = 5$, AAV6-*Caren*: $n = 6$). Levels seen in the AAV6-*GFP* group were set to 1. **b** Representative western blots (left) and quantitation (right) of Hint1 protein levels in heart tissues of WT mice injected with $1 \times 10^{11}$ vg per mouse of AAV6-*GFP* or AAV6-*Caren*, 1 week after injection (AAV6-*GFP*: $n = 5$, AAV6-*Caren*: $n = 6$). Hsc70 served as the loading control. Protein intensity values in the AAV6-*GFP* group were set to 1. **c** Schematic showing experimental protocol. **d** Comparison of %FS between mice before (pre) and 1 week after TAC surgery ($n = 33$) (left) and in TAC mice injected with PBS or $1 \times 10^{11}$ vg per mouse of AAV6-*GFP* or AAV6-*Caren* (sham: $n = 5$, PBS: $n = 12$, AAV6-*GFP*: $n = 11$, AAV6-*Caren*: $n = 10$) (right). **e** Representative M-mode echocardiography recordings in sham-operated and TAC-operated animals 7 weeks after injection of PBS, AAV6-*GFP*, or AAV6-*Caren*. **f** Left ventricular end-diastolic diameter (LVDd) (mm) (left), left ventricular end-systolic diameter (LVDs) (mm) (middle), and percent fractional shortening (%FS) (right) (sham: $n = 5$, PBS: $n = 12$, AAV6-*GFP*: $n = 11$, AAV6-*Caren*: $n = 10$). Box plots for **a** and **f** present min to max, median, and all points. Data show mean ± SD and all points for **b**. For **d**, data show mean ± SD. Statistical significance was determined by two-sided unpaired Student's *t*-test (**a**, **b**) or one-way ANOVA with Sidak's post hoc test (**d**, **f**). In **d**, the *p*-value show AAV6-*GFP* vs. AAV6-*Caren*. †$p < 0.0001$, n.s; not significant, between groups.

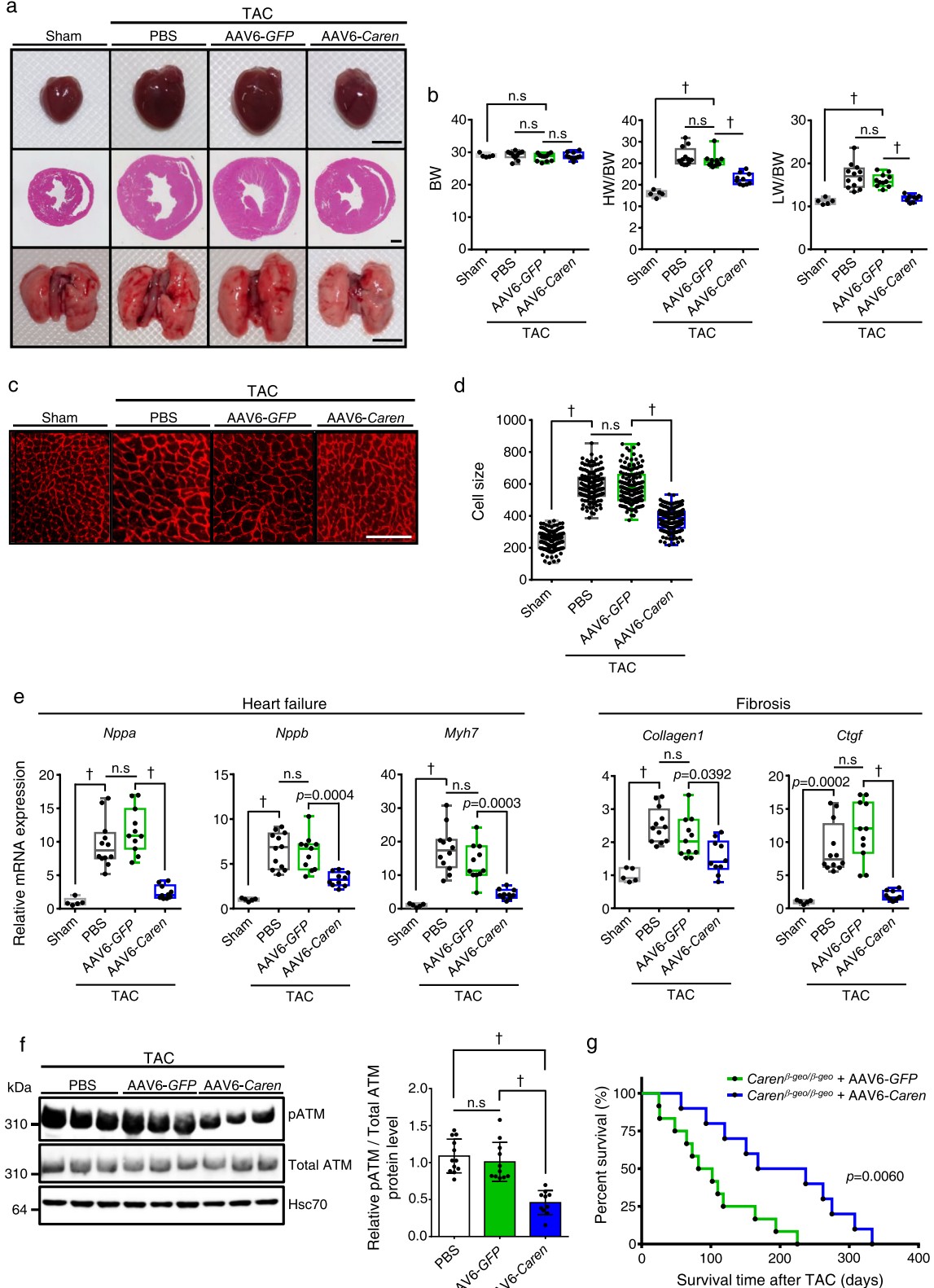

U/ml RNase inhibitor, pH 7.4) for 10 min on ice and treated with DNase (TURBO DNase, Thermo Fisher Scientific Inc) for 10 min at 37 °C. DNase-treated samples were then mixed with 1/10 volume of 10% N-lauroylsarcosine and sonicated to obtain whole-cell lysates.

Samples and 1.4× hybridization buffer (28 mM Tris-HCl, 9.8 mM EDTA, 4.2 mM EGTA, 210 mM LiCl, 1.4% NP-40, 0.28% N-lauroylsarcosine, 0.14% sodium deoxycholate, 4.2 M guanidine thiocyanate, 3.5 mM TCEP, pH 7.5) were mixed at a ratio of 1: 2.5 and centrifuged at 16,000 × g for 10 min at 4 °C.

Supernatants were incubated with streptavidin magnetic beads (MyONE Streptavidin C1 magnetic beads, Thermo Fisher Scientific Inc.) for 20 min at 37 °C. After incubation, streptavidin-cleared samples were collected and incubated with 100 pmol of 5'-biotinylated antisense single-strand DNA (ssDNA) probes against *Hint1* or *LacZ* (a negative control) mRNA (Supplementary Table 5) for 3 h at 37 °C. After hybridization, samples were incubated with streptavidin magnetic beads for 30 min at 37 °C and washed 6 times in wash buffer (20 mM Tris-HCl, 10 mM EDTA, 1% NP-40, 0.2% N-lauroylsarcosine, 0.1% sodium deoxycholate, 3 M

**Fig. 9 Intravenous injection of AAV6-*Caren* suppresses myocardial remodeling in mice subjected to pressure overload. a** Gross appearance of whole heart (top row), HE-stained sections of the mid-portion of the heart (middle row; scale bar, 1 mm), and gross appearance of the whole lung (bottom row; scale bar, 5 mm). **b** Body weight (BW) (g) (left), HW/BW ratio (mg/g) (middle), and lung weight per body weight ratio (LW/BW) (mg/g) (right) (sham: $n = 5$, PBS: $n = 12$, AAV6-*GFP*: $n = 11$, AAV6-*Caren*: $n = 10$). **c** Shown are representative left ventricle sections stained with wheat germ agglutinin (WGA) to indicate cardiomyocyte size (scale bar, 100 μm). **d** Distribution of sizes of myocardial cells (μm$^2$) in indicated mice (sham: $n = 145$ cells, PBS: $n = 149$ cells, AAV6-*GFP*: $n = 149$ cells, AAV6-*Caren*: $n = 150$ cells). **e** Relative expression of genes associated with heart failure and fibrosis in hearts (sham: $n = 5$, PBS: $n = 12$, AAV6-*GFP*: $n = 11$, AAV6-*Caren*: $n = 10$). Levels seen in sham-operated WT mice were set to 1. **f** Representative western blot (left) and pATM/total ATM ratio (right) in heart tissues of TAC mice injected with PBS, AAV6-*GFP*, or AAV6-*Caren*, 8 weeks after TAC operation (PBS: $n = 12$, AAV6-*GFP*: $n = 11$, AAV6-*Caren*: $n = 10$). Hsc70 served as a loading control. Values derived from the PBS group were set to 1. **g** Kaplan-Meier curve showing mortality of TAC-operated *Caren*$^{β-geo/β-geo}$ mice injected with $1 \times 10^{11}$ vg per mouse of AAV6-*GFP* ($n = 12$) or AAV6-*Caren* ($n = 10$). Box plots for **b**, **d**, and **e** present min to max, median, and all points. For **f**, data show mean ± SD and all points. Statistical significance was determined by one-way ANOVA with Sidak's post hoc test (**b**, **d**, **e**, **f**) or log lank test (**g**). $^†p < 0.0001$, n.s; not significant, between groups.

guanidine thiocyanate, 2.5 mM TCEP, pH 7.5) for 5 min at 45 °C. Beads were washed with elution buffer (50 mM Tris-HCl, 75 mM NaCl, 3 mM MgCl$_2$, 0.125% N-lauroylsarcosine, 0.025% sodium deoxycholate, 2.5 mM TCEP, pH 7.5), resuspended in 55 μl elution buffer, treated with 7.5 μl RNase H (New England Biolabs, Ipswich, MA, USA), and then incubated 30 min at 37 °C. After RNase H digestion, supernatants were collected as a first eluate. Beads were resuspended in 62.5 μl 1× hybridization buffer and incubated for 5 min at 37 °C. Supernatants were then collected and combined with the first eluates.

For reverse crosslinking and protein digestion, pooled eluates were mixed with 312.5 μl digestion buffer (20 mM Tris-HCl, 10 mM EDTA, 2% N-lauroylsarcosine, 2.5 mM TCEP, pH 7.5), 50 μl 5 M NaCl, and 5 μl Proteinase K (Sigma-Aldrich) and incubated 1 h at 65 °C. RNAs were extracted using a Zymo RNA Clean and Concentrator-5 kit (Zymo Research, Irvine, CA, USA), according to the manufacturer's instructions. RNA samples were treated with DNase (TURBO DNase, Thermo Fisher Scientific Inc) and Exonuclease I (New England Biolabs), and purified using a Zymo RNA Clean and Concentrator-5 kit (Zymo Research). RNAs were then subjected to real-time quantitative RT-PCR using primer pairs specific for *Caren* and *Hint1* (Supplementary Table 5). In the RAP protocol using ssDNA probes against *Hint1* mRNA, the average yield of *Hint1* mRNA was approximately 10% and the amount of purified *Hint1* mRNA in the RAP protocol using ssDNA probes against *Hint1* mRNA was more than 10,000-fold higher than that seen in the RAP protocol using ssDNA probes against *LacZ* mRNA.

**Isolation of mouse heart mitochondria.** Mouse heart tissue was homogenized in mitochondrial isolation buffer (210 mM mannitol, 70 mM sucrose, 5 mM HEPES-KOH, 1 mM EGTA, 0.5% fatty acid-free bovine serum albumin (BSA), pH 7.2) using a glass-Teflon homogenizer. Homogenates were centrifuged at 27,000 × g for 10 min at 4 °C, and pellets suspended in mitochondrial isolation buffer. Samples were centrifuged at 500 × g for 5 min at 4 °C, and the supernatant was collected and centrifuged at 10,000 × g, 4 °C for 5 min. The pellet was then resuspended in mitochondrial isolation buffer, and samples centrifuged at 10,000 × g for 5 min at 4 °C. The resulting pellet was suspended in BSA-free mitochondrial isolation buffer, and mitochondrial protein concentration determined using the Bio-Rad Protein Assay Dye Reagent (Bio-Rad Laboratories, Hercules, CA, USA). Samples were then incubated with an added equal volume of mitochondrial isolation buffer and either assayed for oxygen consumption and mitochondrial complex activity or subjected to blue native (BN)-PAGE analysis.

**Oxygen consumption analysis of isolated mitochondria.** Isolated mitochondria were diluted with mitochondrial assay solution (220 mM mannitol, 70 mM sucrose, 2 mM HEPES-KOH, 1 mM EGTA, 10 mM KH$_2$PO$_4$, 5 mM MgCl$_2$, 0.2% fatty acid-free BSA, 2 mM malate, 10 mM pyruvate, pH 7.2) to 80 μg/ml. Diluted mitochondria were loaded at 50 μl per well into 24-well Seahorse assay plates (Seahorse Bioscience) and centrifuged at 2000 × g for 20 min at 4 °C. After centrifugation, 450 μl mitochondrial assay solution was added to wells, and plates were incubated for 8 min at 37 °C. To assess mitochondrial respiratory function, the following compounds (final concentrations) were sequentially injected into each well: ADP (4 mM), oligomycin (2 μM), FCCP (4 μM), and Antimycin A (4 μM). OCR was measured under basal conditions and after each injection using an XFe24 extracellular flux analyzer (Seahorse Bioscience).

**Analysis of mitochondrial respiratory chain complex levels.** To examine levels of each mitochondrial respiratory chain complex, 125 μg isolated mitochondria were suspended in 40 μl solubilizing buffer (50 mM Bis-Tris, 1 M 6-aminocaproic acid, protease inhibitors) and incubated with 6 μl 10% DDM (n-dodecyl β-D-maltoside). Samples were centrifuged at 100,000 × g for 15 min at 4 °C. The supernatant was mixed with 3 μl 5% Serva G (5% brilliant blue G, 1 M 6-aminocaproic acid) and served as a mitochondrial protein sample. Ten microliters of the sample were subjected to blue native-PAGE (BN-PAGE) analysis using NativePAGE Novex Bis-Tris Gel System (Thermo Fisher Scientific Inc), according to the manufacturer's instructions.

For immunoblotting analysis of each mitochondrial respiratory chain complex, proteins were separated by BN-PAGE and transferred to PVDF membranes. Membranes were incubated with anti-NDUFB8 (Complex I, ab110242), anti-SDHB (Complex II, ab14714), anti-UQCRCI (Complex III, ab110252), anti-MTOCI (Complex IV, ab14705), and anti-ATP5A (Complex V, ab14748). All antibodies were purchased from Abcam. Antibodies were diluted at 1:2000, and samples were incubated at 4 °C overnight. After PBST washing, membranes were incubated with 1:3000 diluted HRP-conjugated sheep anti-mouse IgG (GE Healthcare Life Science, Piscataway, NJ, USA) antibodies at room temperature for 60 min.

**Measurement of mitochondrial respiratory chain complex activity.** Activities of complexes I, II/III, IV, and V in isolated mouse heart mitochondria were measured using a MitoCheck Complex I Activity Assay Kit, a MitoCheck Complex II/III Activity Assay Kit, a MitoCheck Complex IV Activity Assay Kit, and a MitoCheck Complex V Activity Assay Kit, all from Cayman Chemical, Ann Arbor, MI, USA, according to the manufacturer's instructions.

**Human samples.** Human studies included 9 patients (4 men, 5 women; mean age ± s.e.m., 67.8 ± 19.9 years, Supplementary Table 1). In all cases, written informed consent for the study of retained tissues from postmortem examination was obtained from closest relatives. The study was also approved by the Ethics Committees of Kumamoto University (approval No. 1454). Human heart tissue samples were fixed 24 h in 4% paraformaldehyde and embedded in paraffin.

For RT-PCR analysis, total RNA was extracted from 3 formalin-fixed paraffin-embedded (FFPE) tissue slides holding sections of 10 μm thickness using an RNeasy FFPE Kit (Qiagen, Valencia, CA, USA), and then quantified and quality-checked using a NanoDrop One C spectrophotometer (Thermo Fisher Scientific, Waltham, MA, USA). For RT-PCR, TaqMan specific primer/probe sets for *Caren*, human *NPPB*, human *18S*, and TaqMan custom primer/probe sets for human lincRNA transcripts 1 and 3 were purchased from Thermo Fisher Scientific Inc (Waltham, MA, USA): Sequences of transcript 1 were forward (5′-AACTATCG AGGCCGCAAATTAA-3′) and reverse (5′-GAACTCGAGAGGCGGAGGTT-3′) primers and probe (5′-TTCTCAACTGTGACAGTTT-3′), and Sequences of transcript 3 were forward (5′-GGAAACTTGGAACCTCACCAATA-3′) and reverse (5′-CGCTTTTCAGGACTCAGGAAAT-3′) primers and probe (5′-CCTG CATTAGCTCTAC-3′).

For immunohistological analysis, embedded paraffin blocks of heart tissue were cut into 4-μm-thick sections, air-dried, and deparaffinized. For immunohistochemistry, sections were pretreated with periodic acid (Nichirei, Tokyo, Japan) to inhibit endogenous peroxidases. Subsequently, specimens were incubated overnight with rabbit polyclonal anti-human HINT1 antibody (1 mg/ml) at 4 °C (Ab124912, Abcam). After washing with PBS, specimens were incubated with 500-fold diluted goat anti-rabbit IgG conjugated with peroxidase (GE Healthcare Life Science) as a second antibody at room temperature for 60 min, and specimens were counterstained with hematoxylin. Peroxidase activity was visualized by incubation with a 3,3-diaminobenzidine solution and analyzed by light microscopy (model BZ-9000; Keyence).

**Analysis of protein-coding potential.** To search for encoded peptides we analyzed the protein-coding potential of lncRNA candidates using the NCBI ORF Finder (http://www.ncbi.nlm.nih.gov/gorf/gorf.html) and the Coding Potential Calculator (http://www.cpc.cbi.pku.edu.cn)[28].

**Statistics and reproducibility.** Sample size and information about statistical tests are reported in the figure legends. No statistical methods were used to determine sample size, but sample sizes were determined based on previous reports[32,33]. No exclusion/inclusion criteria were applied to mice used in this study. Group allocation and outcome assessment were performed in a blinded manner. In vitro experiments were repeated at least three times with similar results. All values were reported as the mean ± SD. Data were assessed with two-group comparisons of

variables using two-sided unpaired Student's *t*-test, and with multiple comparisons by one-way ANOVA with Tukey's multiple comparisons test between each group. Mouse survival was analyzed by the Kaplan–Meier log-rank test. Analysis of Kaplan–Meier and ANOVA data was performed using GraphPad Prism software (version 7.03, GraphPad Software). For human studies, Spearman's test and correlation coefficients were used to evaluate relationships between lincRNA transcript 1 or 3 expression and *NPPB* levels using STATA 15.0 software (Stata Corp. LLC., Lakeway Drive College Station, TX, USA). For all statistical analyses, a value of $p < 0.05$ was considered statistically significant.

**Reporting summary**. Further information on research design is available in the Nature Research Reporting Summary linked to this article.

## Data availability
The proteomic analysis data presented in Figs. 4a and 5a have been deposited to the ProteomeXchange Consortium via the PRIDE partner repository (https://www.ebi.ac.uk/pride/)[68] with an accession number P6. The following publicly available databases were used: STRING database (https://string-db.org) for performing enrichment analysis of differentially expressed proteins, and ENCODE database (http://genome.ucsc.edu/ENCODE/index.html) for analyzing histone modifications of interest lincRNA genes and obtaining sequences of human lincRNA transcripts. All remaining data supporting the findings of this study are available within the article and its Supplementary Information files or from the corresponding author upon reasonable request. Source data are provided with this paper.

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

## Acknowledgements
We thank our colleagues for valuable suggestions and discussion. We also thank K. Tabu, Y. Ito, S. Iwaki, T. Nakamura, N. Shirai, M. Kamada, H. Ishiguro, Y. Einaga, S. Kurosaki, and K. Igata for technical assistance. This work was supported by the Scientific Research Fund of the Ministry of Education, Culture, Sports, Science and Technology (MEXT) of Japan (Grant 18H02809 to Y.O., Grant 18K19519 to Y.O., Grant 19K17607 to M.S., Grant 25461114 to K.M., Grant 19K08560 to K.M. and Grant 23310135 to K.A.), the Core Research for Evolutional Science and Technology (CREST) Program of the Japan Science and Technology Agency (JST) (Grant 13417915 to Y.O.), the CREST Program of the Japan Agency for Medical Research and Development (AMED) (Grant 18gm0610007 to Y.O.), the Project for Elucidating and Controlling Mechanisms of Aging and Longevity (AMED) (Grant 19gm5010002 to Y.O.), the Takeda Science Foundation (Y.O. and M.S.), the SENSHIN Medical Research Foundation (Y.O. and M.S.), the Grant for Basic Research of the Japanese Circulation Society (2020) (M.S.), the Harold S. Geneen Charitable Trust Awards Program for Coronary Heart Disease Research (J.S.W.), the Nora Eccles Treadwell Foundation (J.S.W., S.F.), and NIH Grant R01HL130424-01 (S.F.).

## Author contributions
M.S., T.K., K.M., J.S.W., and Y.O. conceived the study and designed experiments. M.S., T. K., K.M., J.S.W., Z.T., S.Z., H.H., A.M., A.B., J.M., T.S., K.H., K.Y., M.I., M.A., Y.K., T.W., S.N., S.F., K.N., and K.A. collected or analyzed data or performed experiments. All authors discussed the results and approved the final version of the manuscript.

## Competing interests
The authors declare no competing interests.
