## [Peer Review File · Nature Communications]

Reviewer #1 (Remarks to the Author):

In this study, Sato et al. demonstrated a novel mouse cardiac-enriched lncRNA Caren has cardioprotective effects. By using four different animal models including Caren knockout, Caren CAG knockin, and AAV6 injection, Caren was proved to maintain cardiac function under pressure overload condition (TAC) by inhibiting Hint1 expression and therefore inactivating ATM-DDR pathway. In general, it is amazing to see so many different animal models were applied to study lncRNA Caren. However, there are some mechanistic questions that need to be answered experimentally to improve this study.

1. Overexpression of the lncRNA Caren by either CAG knockin or AAV6 injection was shown to be cardioprotective. The authors tried to apply the use of Caren as therapeutic target for treating heart failure patients in the future. However, there are no experiments, such as RNA pulldown or RNA IP, to prove the direct relationship or interaction between Caren and Hint1. To make lncRNA as therapeutic target clinically, I think the detail mechanisms are needed.
2. Continue with question 1, it is good that mouse Caren has a human locus-conserved transcript that expresses in the human iPSC-derived cardiomyocytes. However, more functional studies for translational medicine are needed, for example, if the expression of human transcripts changed under stress conditions such as hypertrophic stimulation (PE, ISO, ET1, etc). What happened to the human iPSC-cardiomyocytes if the human transcripts are inhibited or overexpressed?
3. The authors showed that Caren inhibits protein translation of Hint1 mRNA, do the authors know more detail mechanisms? Does the Caren inhibit Hint1 translation by direct binding? Or by other RNA binding protein? Or the Caren inhibit the ribosome binding? More experimental work is needed here.
4. The authors deleted different regions of Caren and concluded that the effects of Caren come from a structural motif rather than a specific sequence. I think the evidence here is not enough to have this conclusion. The authors could produce Caren by in vitro transcription and either fold it into normal secondary structure or denature it, and then study the functions under these two different RNA structures.

Reviewer #2 (Remarks to the Author):

In this manuscript, Sato et al. examined the role of a novel lncRNA Caren in the pathological development of heart failure. The data are particularly interesting in the suggestion that Caren acts in a cytoplasmic role to impact the DNA damage response and mitochondrial function. Overall, the study is fairly thorough with the generation of a number of genetically-modified models and a focus on manipulation of the axes/proteins associated with Caren. With that being said some concerns exist, which are indicated below.

Of concern is the limited number of indices being reported to reflect cardiac dysfunction. M-mode echocardiography is utilized but only one functional parameter is being reported (%FS). The authors could make a stronger case with inclusion of more functional parameters (i.e. %EF, CO, SV, heart rate, etc...). The authors are using the Vevo2100, so those parameters should be generated following an M-mode trace. The inclusion of strain dynamics would also have been helpful if B-mode imaging was performed. Also, on a side note, please include the anesthesia methodology for the

echocardiography.

The authors try to make a case for the Caren AAV-6 (Fig. 7) attenuating the development of cardiac dysfunction, but the echo trace for the Caren AAV-6 is not very convincing. Further, the inclusion of only %FS does not make a very strong case (see comment above). As above, expanding the reported parameters obtained by M-mode would be beneficial.

The authors did a very thorough proteomic analysis of whole tissue lysates of ventricles obtained from Caren β -geo/ β -geo, CAG-Caren TG, and WT littermate mice without undergoing TAC to show differences in ventricle protein expression contributed to the level of Caren expression at baseline. While protein expression analysis is informative for baseline protein levels, it does not necessarily mean there are differences in mitochondrial function. To determine if the increased protein expression in pathways of mitochondria biogenesis and OXPHOS is due to improved intrinsic mitochondrial function and/or due to an increased number of mitochondria, the authors will need to make evaluations in cardiac myocytes or isolated mitochondria from Caren β -geo/ β -geo, CAG-Caren TG, and WT littermate mice.

If the authors insist that the gene expression profile of mitochondrial biogenesis and OXPHOS is improved in CAG-Caren Tg mice compared to Caren β -geo/ β -geo and WT littermate mice due to downregulation of Hint1 by Caren (at baseline without TAC), why is the gene expression profile in Hint1 \pm not restored to levels comparable to WT littermate mice (at baseline without TAC)? This observation implies that the contribution of Caren expression is not solely responsible for the improved gene expression profile of mitochondrial biogenesis and OXPHOS at baseline without TAC. The authors should at least hypothesize potential mechanisms by which increased Caren improves the gene expression profile of mitochondrial biogenesis and OXPHOS at baseline.

Fig. 4e and f; though these data are interesting, they are somewhat incomplete. As the authors know, these complexes are composed of multiple protein components, thus the inclusion of a single protein as surrogate for the complex is not convincing. The authors should consider the inclusion of the entire complex, which can be performed by BN-PAGE. I would also include enzymatic activities for the individual ETC complexes. This would strengthen the hypotheses being posited.

Fig. 5k and l; the authors report Seahorse analyses, but exclude some parameters, including proton leak. Since this data is generated by the analyses, their inclusion should be added and may help to support their conclusions.

Fig. 1h; the authors show subcellular differences in several lncRNAs including Caren. The data are interesting but difficult to interpret without the inclusion of lncRNAs that have been established in a given subcellular region (i.e. nuclear or cytosol). The inclusion of a lncRNA known to exist primarily in a given subcellular (i.e. Xist or Hotair) should be included for reference.

Supplementary Fig. 1d and e; the authors make an effort to correlate between mouse models for Caren expression by inclusion of analyses in the chromosomal region that is similarly arrayed. The data yield a number of potential sequences that may be the human orthologue for Caren. Two sequences are selected but it is unclear whether either is the human Caren transcript. Inclusion of data from human patients (preferably heart failure) for changes in the expression of either of these transcripts as well as proteomic evaluation of Hint1, would enhance the translatability of the data.

Why was the Hint1^{+/-} mouse subjected to TAC instead of the Hint1^{-/-} mouse?

Reviewer #3 (Remarks to the Author):

In this manuscript, the authors discovered a novel lncRNA by a gene trap approach in mouse ES cells. The authors found the lncRNA (Caren) is expressed in the heart and then analyzed its functions. By knocking out the gene and ectopic expression of the gene (transgenic experiment and by using AAV virus), the authors found that Caren has the function of anti-HF activity. Then the authors characterized the target of Caren by proteomic analysis, and identified mitochondrial biogenesis and Hint1. Finally, the author found a link between Caren and the ATM-DDR pathway. The genetic experiments described in the manuscript were well-designed and the results obtained by the authors are very clear.

Comments: The authors tried to dissect Caren by constructing transgenic mice and concluded that the entire structure of Caren may be important. They can perform experiments using the in vitro translation system of Hint1 to see if similar results can be obtained. Also, there should be a target sequence on Hint1 RNA if their hypothesis is correct. I like to see the results of these experiments.

Reviewer #4 (Remarks to the Author):

Title: The lncRNA Caren protects against heart failure by inactivating the ATM-DNA damage response pathway and activating mitochondrial biogenesis

Summary: The authors describe a novel cytoplasmic lncRNA, named Caren, which plays a role in protecting the heart against heart failure caused by pathological stress. The authors propose a model where Caren in cardiomyocytes prevents the translation of Hint1 and thereby hinder the full activation of ATM upon transverse aortic constriction (TAC). The manuscript is written extremely well, very clearly and the conclusions are supported by experimental data with the exception of one point, please see the following section.

Major points:

1) As reported by the authors, the loss of Caren leads to multiple changes in transcript levels and protein concentrations. They have not shown that in the mice that have lost Caren it is specifically the over activation of ATM upon TAC that causes heart failure, meaning they have not demonstrated a causal effect for their model in their system. My question is, therefore, in the beta-geo/beta-geo mice do the authors see a difference in the activation of ATM compared to wild-type mice as you would expect there to be a stronger activation upon TAC. Furthermore, would inhibiting ATM activity by an ATM inhibitor in beta-geo/beta-geo mice correct for the lower survival presented in Fig. 2i upon TAC.

Minor points:

1) Please define "HF" upon first use in the text.
2) Fig. 3j, Fig. 6h, Fig. 7l and supplementary Fig. 4h. Please add the Western blot for unphosphorylated ATM as a control that Caren does not influence the amount of total ATM. This will support the conclusion that Caren prevents ATM from becoming fully activated due to the suppression of Hint1 translation and not due to any effect that Caren may have on the amount of

ATM protein in the cell.

3) In the discussion section the authors state that “However, Hint1 reportedly functions as a tumor suppressor by activating ATM and the subsequent DDR.” This statement is not completely correct. Hint1 does not activate ATM, it contributes to the activation of ATM upon DNA damage. So it is the DNA damage that activates ATM and Hint1 helps during this process. I realize that the reason for DNA damage dependent ATM activation upon TAC is not known and well beyond the scope of this paper, but I would please ask the authors to clarify this statement and maybe add a line or two as to how TAC could theoretically cause DNA damage that would activate the DNA damage response kinase ATM.

Responses to comments by Reviewer 1

Thank you for constructive comments, which were extremely helpful. We have extensively revised the manuscript based on those suggestions and feel that our manuscript is much improved. Our answers are as follows.

1. Overexpression of the lncRNA Caren by either CAG knockin or AAV6 injection was shown to be cardioprotective. The authors tried to apply the use of Caren as therapeutic target for treating heart failure patients in the future. However, there are no experiments, such as RNA pulldown or RNA IP, to prove the direct relationship or interaction between Caren and Hint1. To make lncRNA as therapeutic target clinically, I think the detail mechanisms are needed.

Based on these suggestions, we employed an RNA pull-down assay to assess potential direct *Caren/Hint1* mRNA interaction (see Figure for reviewer only below). When biotinylated control or *Caren* RNA was incubated with lysates of mouse myoblast C2C12 cells, *Hint1* mRNA abundantly bind to *Caren* RNA relative to control RNA. However, when biotinylated *Caren* RNA was incubated with total RNA isolated from C2C12 cells, levels of *Hint1* mRNA bound to *Caren* markedly decreased, suggesting that *Caren* and *Hint1* mRNA interaction is indirect. Some lncRNAs reportedly regulate target mRNA translation by binding to RNA-binding proteins or ribosomes (Rashid *et al.*, *Genomics Proteomics Bioinformatics*, 14: 73-80, 2016; Yao *et al.*, *Nat Cell Biol*, 21: 542-551, 2019). Thus, the *Caren/Hint1* mRNA interaction may also be mediated by RNA-binding proteins or ribosomes.

We agree that elucidating molecular mechanisms underlying *Caren* regulation of *Hint1* mRNA is necessary to develop therapeutic strategies against heart failure. However, we feel that our finding that *Caren* indirectly interacts with *Hint1* mRNA sheds light on these mechanisms. In future analyses we will focus on identifying molecules functioning in this interaction.

Figure Caren indirectly interacts with Hint1 mRNA. Cell lysates or total RNA from C2C12 cells were incubated with biotinylated control RNA (5'-CCUGGUUUUUUAAGGAGUGUCGCCAGAGUGCCGCGAAUGAAAAA-3') or *Caren* transcripts. Biotinylated RNAs were then collected using streptavidin magnetic beads, and RNAs bound to control or *Caren* RNAs were isolated and subjected to RT-PCR analysis for *Hint1* mRNA (n = 2 per group). Levels in *Caren*-bound RNAs from cell lysate were set to 1.

2. Continue with question 1, it is good that mouse *Caren* has a human locus-conserved transcript that expresses in the human iPSC-derived cardiomyocytes. However, more functional studies for translational medicine are needed, for example, if the expression of human transcripts changed under stress conditions such as hypertrophic stimulation (PE, ISO, ET1, etc). What happened to the human iPSC-cardiomyocytes if the human transcripts are inhibited or overexpressed?

Based on comment #7 by Reviewer #2, we examined expression of human lincRNA transcripts 1 and 3 in human heart tissues obtained from autopsy cases and found that expression levels of lincRNA transcript 1, but not transcript 3, were inversely correlated with *BNP* mRNA (a marker of left ventricular dysfunction) and HINT1 protein expression (Supplementary Fig. 7a–c).

We then analyzed lincRNA transcript 1 using human iPSC cell-derived cardiomyocytes (iPSC-CMs), as you suggest. Treatment of human iPSC-CMs with isoproterenol (Iso) decreased lincRNA transcript 1 relative to those seen in control cells (Supplementary Fig. 7d). We also knocked down expression of human lincRNA transcript 1 in human iPSC-CMs using two different siRNAs

(siLincRT-A and siLincRT-B) and analyzed mitochondrial respiratory capacity relative to negative control siRNA (siScramble) cells (Supplementary Fig. 7e). Cell Mito Stress test showed that basal and maximal respiration levels and ATP production in human lincRNA transcript 1-knockdown iPSC-CMs were significantly decreased relative to control cells (Supplementary Fig. 7f).

Moreover, our new data demonstrated that respiratory capacity of mitochondria isolated from heart tissues of *Caren* ^{β -geo/ β -geo} mice was significantly decreased relative to that seen in WT littermate mice (Supplementary Fig. 4f–h).

Overall, we feel that human lincRNA transcript 1 is a candidate *Caren* homolog, although further studies are required to confirm this function. We now report these findings in the Results section on Page 20 Lines 11–21, and discuss these issues on Page 25 Lines 4–7.

3. The authors showed that Caren inhibits protein translation of Hint1 mRNA, do the authors know more detail mechanisms? Does the Caren inhibit Hint1 translation by direct binding? Or by other RNA binding protein? Or the Caren inhibit the ribosome binding? More experimental work is needed here.

These comments overlap somewhat with your comment #1. As we state there, our data suggest that *Caren* interacts with *Hint1* mRNA indirectly, and possibly inhibits its translation by binding RNA-binding proteins or ribosomes. We plan to identify these molecules in the future.

4. The authors deleted different regions of Caren and concluded that the effects of Caren come from a structural motif rather than a specific sequence. I think the evidence here is not enough to have this conclusion. The authors could produce Caren by in vitro transcription and either fold it into normal secondary structure or denature it, and then study the functions under these two different RNA structures.

Thank you for these comments. Based on this suggestion, we assessed *Hint1* mRNA translation *in vitro* in the presence of either native *Caren* or *Caren* that had been *in vitro* transcribed and then denatured by a snap cold RNA denaturing method (Novikova *et al.*, *Nucleic Acids Res* 40: 5034-5051, 2012). As

anticipated, the presence of native *Caren* significantly decreased Hint1 protein levels compared to controls, whereas in the presence of denatured *Caren* we observed no effects on *Hint1* mRNA translation (Supplementary Fig. 4k). These results suggest that native secondary structure is required for *Caren* to regulate *Hint1* mRNA translation. We report these findings in the Results section on Page 13 Lines 14–16.

Responses to comments by Reviewer 2

Thank you for constructive comments, which were extremely helpful. We have extensively revised the manuscript based on those suggestions and feel that our manuscript is much improved. Our answers are as follows.

1. Of concern is the limited number of indices being reported to reflect cardiac dysfunction. M-mode echocardiography is utilized but only one functional parameter is being reported (%FS). The authors could make a stronger case with inclusion of more functional parameters (i.e. %EF, CO, SV, heart rate, etc...). The authors are using the Vevo2100, so those parameters should be generated following an M-mode trace. The inclusion of strain dynamics would also have been helpful if B-mode imaging was performed. Also, on a side note, please include the anesthesia methodology for the echocardiography.

The authors try to make a case for the Caren AAV-6 (Fig. 7) attenuating the development of cardiac dysfunction, but the echo trace for the Caren AAV-6 is not very convincing. Further, the inclusion of only %FS does not make a very strong case (see comment above). As above, expanding the reported parameters obtained by M-mode would be beneficial.

Based on these suggestions, we added the following functional parameters calculated from M-mode measurements in the Fig. 2, 3, 6, 7, and Supplementary Fig. 3, 5, and 6: left ventricular end-diastolic volume (LVEDV), left ventricular end-systolic volume (LVESV), cardiac output (CO), percent ejection fraction (%EF), stroke volume (SV), and heart rate (HR) (Supplementary Table 1). However, because we did not perform B-mode imaging and lacked the software (Vevostrain software) required to analyze strain dynamics, we could not show strain dynamics data.

Overall, our additional analysis demonstrates that TAC mice injected with AAV6-GFP or PBS showed comparable left ventricular dilatation with reduced fractional shortening, whereas mice treated with AAV6-Caren showed attenuated development of cardiac dysfunction and left ventricular dilatation relative to controls (Fig. 7c-f and Supplementary Table 1). Moreover, in the revision, we found that LVEF, LVEDV, LVESV, SV, and CO were maintained in mice treated with AAV6-Caren compared with control mice treated with GFP or PBS (Supplementary Table 1), supporting our conclusion that Caren re-expression in failing heart slows heart failure progression.

Finally, we have added anesthesia methodology used for echocardiography to the Materials & Methods section (Page 27 Lines 4–12).

2. *The authors did a very thorough proteomic analysis of whole tissue lysates of ventricles obtained from Carenb-geo/b-geo, CAG-Caren TG, and WT littermate mice without undergoing TAC to show differences in ventricle protein expression contributed to the level of Caren expression at baseline. While protein expression analysis is informative for baseline protein levels, it does not necessarily mean there are differences in mitochondrial function. To determine if the increased protein expression in pathways of mitochondria biogenesis and OXPHOS is due to improved intrinsic mitochondrial function and/or due to an increased number of mitochondria, the authors will need to make evaluations in cardiac myocytes or isolated mitochondria from Carenb-geo/b-geo, CAG-Caren TG, and WT littermate mice.*

Based on your suggestion, we performed extracellular flux analysis using mitochondria isolated from heart tissues of *Caren* ^{β -geo/ β -geo}, CAG-Caren Tg, and WT littermate mice to evaluate cardiac mitochondrial function. Basal respiration, phosphorylating respiration in the presence of ADP (state 3), resting respiration in the presence of oligomycin (state 4o), maximal uncoupling respiration in the presence of FCCP (state 3u), and respiration in the presence of antimycin A (Anti A) were comparable in CAG-Caren Tg and WT littermate mice (Supplementary Fig. 4a, b), as was the respiratory control ratio (RCR, state 3/state 4o) (Supplementary Fig. 4c), suggesting that *Caren* overexpression does not alter mitochondrial respiratory capacity at basal levels in cardiomyocytes. However, we observed a significant decrease in basal, state 3, and state 3u respiration in *Caren* ^{β -geo/ β -geo} compared to WT littermate mice (Supplementary Fig. 4f, g). The RCR in *Caren* ^{β -geo/ β -geo} mice also decreased compared to that seen in WT littermate mice, although the difference was not significant (Supplementary Fig. 4h). These results suggest that mitochondrial function is impaired in *Caren* ^{β -geo/ β -geo} mice. Because these mice show increased cardiac Hint1 levels (Fig. 5b) and increased Hint1 levels reduce mitochondrial respiratory capacity in cardiomyocytes (Fig. 5k, l), impaired mitochondrial function seen here could be due to increased Hint1 levels in cardiomyocytes.

Based on your comment #3, we examined levels of mitochondrial respiratory chain complexes in mitochondria isolated from heart of CAG-Caren Tg and WT littermate mice using BN-PAGE analysis (Supplementary Fig. 4d). BN-PAGE followed by immunoblotting analysis indicated that overexpression of Caren does not alter levels of mitochondrial respiratory chain complexes. Taken together, these findings suggest that increased protein expression in mitochondrial biogenesis and OXPHOS pathways in the heart of CAG-Caren Tg mice is due to an increased number of mitochondria. We report these findings in the Results section on Page 11 Line 11, through Page 12, Line 21.

3. If the authors insist that the gene expression profile of mitochondrial biogenesis and OXPHOS is improved in CAG-Caren Tg mice compared to Caren β -geo/ β -geo and WT littermate mice due to downregulation of Hint1 by Caren (at baseline without TAC), why is the gene expression profile in Hint1 \pm not restored to levels comparable to WT littermate mice (at baseline without TAC)? This observation implies that the contribution of Caren expression is not solely responsible for the improved gene expression profile of mitochondrial biogenesis and OXPHOS at baseline without TAC. The authors should at least hypothesize potential mechanisms by which increased Caren improves the gene expression profile of mitochondrial biogenesis and OXPHOS at baseline.

As you note, CAG-Caren Tg mice exhibit increased expression of genes functioning in mitochondrial biogenesis and OXPHOS, phenotypes not seen in the Hint1 \pm heart. In addition, we demonstrated that mice injected with AAV6-Caren also show increased expression of those genes (Supplementary Fig. 6i, j). On the other hand, Hint1-overexpressing H9c2 cells (Fig. 5k, l) and heart mitochondria of Caren β -geo/ β -geo mice (Supplementary Fig. 4f-h) exhibit decreased mitochondrial respiratory function. These findings suggest that increased Hint1 expression alters mitochondrial respiratory function rather than expression of genes functioning in mitochondrial biogenesis and OXPHOS. Taken together, these results suggest that Caren regulates genes/proteins involved in cardiac mitochondrial biogenesis and OXPHOS independent of Hint1 activity.

PGC-1 α , NRF1/2, TFAM, and ERR α reportedly play central roles in regulating expression of genes functioning in mitochondrial biogenesis and

OXPPOS (Gerald *et al.*, *Genes Dev* 29: 1981-1991, 2015; Gureev *et al.*, *Front Genet* 10: 435, 2019). Importantly, we found that *Nrf2* and *Tfam* expression in hearts of CAG-*Caren* Tg mice was significantly increased relative to hearts of WT littermate mice (Fig. 4c), suggesting that *Caren* overexpression promotes expression of genes functioning in mitochondrial biogenesis and OXPPOS by increasing *Nrf2* and *Tfam* expression. Since *Caren* regulates Hint1 translation, it may also regulate translation of factors that regulate *Nrf2* and *Tfam* transcriptionally. We now discuss these issues on Page 23 Lines 12–22.

4. Fig. 4e and f; though these data are interesting, they are somewhat incomplete. As the authors know, these complexes are composed of multiple protein components, thus the inclusion of a single protein as surrogate for the complex is not convincing. The authors should consider the inclusion of the entire complex, which can be performed by BN-PAGE. I would also include enzymatic activities for the individual ETC complexes. This would strengthen the hypotheses being posited.

Based on this suggestion, we examined levels of individual mitochondrial respiratory chain complexes and their activities in mitochondria isolated from heart tissues of CAG-*Caren* Tg and WT littermate mice (Supplementary Fig. 4d, e). BN-PAGE followed by immunoblotting analysis revealed comparable levels of mitochondrial respiratory chain complexes are comparable between genotypes (Supplementary Fig. 4d). Moreover, we observed no difference in activities of each mitochondrial respiratory chain complex between CAG-*Caren* Tg and WT littermate mice (Supplementary Fig. 4e), suggesting that increased expression levels of protein components of these complexes in the heart of CAG-*Caren* Tg mice is due to an increased number of mitochondria.

Actually, mitochondrial respiratory function in isolated cardiac mitochondria was comparable in CAG-*Caren* Tg and WT littermate mice (Supplementary Fig. 4a–c). In contrast, *Caren* ^{β -geo/ β -geo} mice showed reduced mitochondrial respiratory capacity (Supplementary Fig. 4f–h). Furthermore, although levels of mitochondrial respiratory chain complex I in heart significantly increased in *Caren* ^{β -geo/ β -geo} compared to WT littermate mice (Supplementary Fig. 4i), complex I activity significantly decreased in *Caren* ^{β -geo/ β -geo} mice compared with WT littermate mice (Supplementary Fig. 4j). Moreover, *Caren* ^{β -geo/ β -geo} mice

exhibited a significant increase in levels of mitochondrial respiratory chain complex III in heart relative to WT littermate mice (Supplementary Fig. 4i), while complex III activity was comparable between genotypes (Supplementary Fig. 4j).

Taken together, these findings suggest that *Caren* counteracts HF development by enhancing mitochondrial biogenesis and maintaining mitochondrial respiratory capacity through suppressing Hint1 expression under pathological stress. Furthermore, levels of mitochondrial respiratory chain complexes I and III in heart of *Caren* ^{β -geo/ β -geo} mice may increase in order to compensate for reduced mitochondrial respiratory function. We now report these findings in the Results section on Page 11 Line 11, through Page 12, Line 21, and discuss these issues on Page 22 Line 21, through Page 23 Line 11.

5. Fig. 5k and l; the authors report Seahorse analyses, but exclude some parameters, including proton leak. Since this data is generated by the analyses, their inclusion should be added and may help to support their conclusions.

Based on your comment, we added three parameters (proton leak, coupling efficiency, and non-mitochondrial respiration) to Fig. 5i of the revision. As reported in the first submission, we observed decreases in basal and maximal respiration levels and in ATP production, as well as decreased spare respiratory capacity in H9c2-Hint1 compared to control cells (Fig. 5k, l). Non-mitochondrial respiration was also significantly decreased in H9c2-Hint1 compared to control cells (Fig. 5l). However, there was no difference in proton leak and coupling efficiency between H9c2-Hint1 and control cells (Fig. 5l). These results suggest that an increase in Hint1 levels causes decreased mitochondrial respiratory capacity in cardiomyocytes. We now report these findings in the Results section on Page 14, Lines 7–13.

6. Fig. 1h; the authors show subcellular differences in several lncRNAs including Caren. The data are interesting but difficult to interpret without the inclusion of lncRNAs that have been established in a given subcellular region (i.e. nuclear or cytosol). The inclusion of a lncRNA known to exist primarily in a given subcellular (i.e. Xist or Hotair) should be included for reference.

Based on this comment examined subcellular distribution of *Hotair* using samples used in Fig. 1h. *Hotair* is reportedly distributed in the nucleus (Gupta *et al. Nature* 464, 1071-1076, 2010; Rinn *et al. Cell* 129, 1311-1323, 2007; Tsai *et al. Science* 329, 689-693, 2010). Unlike cytoplasmic *Caren*, *Hotair* was localized to the nucleus, as expected. Because we used heart tissues from male mice for analysis we did not employ *Xist* as a reference. We now report this finding in the Results section on Page 6, Lines 19–22.

7. Supplementary Fig. 1d and e; the authors make an effort to correlate between mouse models for Caren expression by inclusion of analyses in the chromosomal region that is similarly arrayed. The data yield a number of potential sequences that may be the human orthologue for Caren. Two sequences are selected but it is unclear whether either is the human Caren transcript. Inclusion of data from human patients (preferably heart failure) for changes in the expression of either of these transcripts as well as proteomic evaluation of Hint, would enhance the translatability of the data.

Based on this suggestion, we examined expression of two human lincRNA transcripts and HINT1 protein in human heart tissues obtained from autopsy cases. Expression of lincRNA transcript 1 was significantly and inversely correlated with that of *BNP*, a marker for left ventricular dysfunction ($\rho = -0.77$, $p = 0.016$) (Supplementary Fig. 7a). We observed no correlation between expression of lincRNA transcript 3 and *BNP* ($\rho = -0.1$, $p = 0.798$) (Supplementary Fig. 7b). Immunohistochemistry revealed abundant HINT1 protein expression in heart tissue from cases 1 and 2 with the low level of lincRNA transcript 1 expression and high *BNP* expression. Conversely, HINT1 levels in heart tissue from cases 3 and 4 showing high levels of lincRNA transcript 1 and low *BNP* expression were lower than those seen in cases 1 and 2 (Supplementary Fig. 7c).

Interestingly, lincRNA transcript 1 expression levels were significantly decreased in isoproterenol-treated human iPSC-CMs relative to untreated cells (Supplementary Fig.7d). Also, knockdown of human lincRNA transcript 1 in iPSC-CMs decreased mitochondrial respiratory capacity relative to controls (Supplementary Fig. 7f, g). Overall, these results suggest that human lincRNA transcript 1 could be a candidate functional human *Caren* homolog, although further studies are needed to confirm this. We now report these findings in the

Results section on Page 19 Line 17, through Page 20 Line 21.

8. Why was the *Hint1*^{+/-} mouse subjected to TAC instead of the *Hint1*^{-/-} mouse?

Hint1^{-/-} mice show significantly decreased body weight compared to WT mice (Su *et al.*, *Proc Natl Acad Sci U S A.* 100: 7824-9, 2003) (see Figure below, unpublished data), while *Hint1*^{+/-} mice do not, although they exhibit a 50% relative decrease in Hint1 protein levels, like CAG-*Ceren* Tg mice. Body weight differences can alter severity of phenotypes observed in TAC-induced heart failure mouse model; thus we used heterozygotes to investigate the impact of decreased Hint1 expression on the progression of heart failure during pressure overload.

Body weight (g) of 3-month-old WT (n = 10), *Hint1*^{+/-} (n = 17), and *Hint1*^{-/-} (n = 5) male mice. Data are means \pm SD. Statistical significance was determined by one-way ANOVA with Sidak's multiple comparison post hoc test. ** $p < 0.01$, † $p < 0.001$, n.s; not significant, between genotypes.

Responses to comments by Reviewer 3

Thank you for constructive comments, which were extremely helpful. We have extensively revised the manuscript based on those suggestions and feel that our manuscript is much improved. Our answers are as follows.

Comments: The authors tried to dissect Caren by constructing transgenic mice and concluded that the entire structure of Caren may be important. They can perform experiments using the in vitro translation system of Hint1 to see if similar results can be obtained. Also, there should be a target sequence on Hint1 RNA if their hypothesis is correct. I like to see the results of these experiments.

Based on this comment, we assessed *Hint1* mRNA translation *in vitro* in the presence of full-length *Caren* or individual *Caren* fragments A–E (Supplementary Fig. 6h). The presence of full-length *Caren* significantly decreased *Hint1* protein levels compared to controls, while none of *Caren* fragments tested had comparable effects. These results suggest that structure exhibited by full-length *Caren* is important to suppress *Hint1* mRNA translation. We now report these findings in the Results section on Page 17, Lines 10–17.

As described in the responses to comments #1 and #3 by Reviewer #1, our findings suggest that *Caren* interacts with *Hint1* mRNA indirectly, and possibly inhibits its translation by binding RNA-binding proteins (RBPs) or ribosomes. Precise identification of target sequences on *Hint1* mRNA requires that we first define the protein that directly binds that mRNA. We plan to identify that protein in future studies, and once we do, we will conduct analysis to define *Hint1* mRNA target sequences.

Responses to comments by Reviewer 4

Thank you for constructive comments, which were extremely helpful. We have extensively revised the manuscript based on those suggestions and feel that our manuscript is much improved. Our answers are as follows.

Major points:

1) *As reported by the authors, the loss of Caren leads to multiple changes in transcript levels and protein concentrations. They have not shown that in the mice that have lost Caren it is specifically the over activation of ATM upon TAC that causes heart failure, meaning they have not demonstrated a causal effect for their model in their system. My question is, therefore, in the beta-geo/beta-geo mice do the authors see a difference in the activation of ATM compared to wild-type mice as you would expect there to be a stronger activation upon TAC. Furthermore, would inhibiting ATM activity by an ATM inhibitor in beta-geo/beta-geo mice correct for the lower survival presented in Fig. 2i upon TAC.*

Based on your suggestions we examined ATM activation in the heart of TAC-operated *Caren* ^{β -geo/ β -geo} and WT littermate mice (Supplementary Fig. 5j). We showed that the pATM/total ATM ratio significantly increased in the heart of TAC-operated *Caren* ^{β -geo/ β -geo} compared to comparable WT littermate mice, suggesting that *Caren* deficiency enhances ATM activation and accelerates heart failure development.

We agree that investigating whether ATM inactivation prolongs survival of TAC-operated *Caren* ^{β -geo/ β -geo} mice is important for elucidating the significance of *Caren*-mediated regulation of the ATM-DDR pathway in the development of HF. However, it would likely require more than two years to assess a potential life-extending effect of an ATM inhibitor. Thus, instead, we asked whether treatment with an ATM inhibitor would ameliorate HF progression in TAC-operated *Caren* ^{β -geo/ β -geo} mice. To do so, we investigated the effects of KU-60019 (Nakada *et al.*, *Circulation* 139: 1237–1239, 2019; Takahashi *et al.*, *JACC Basic Transl Sci* 4: 234-247, 2019) treatment on cardiac function, heart and lung weight, and ATM activation in TAC-operated *Caren* ^{β -geo/ β -geo} mice (Supplementary Fig. 5k–n). Starting 2 days after TAC surgery, *Caren* ^{β -geo/ β -geo} mice were intraperitoneally administered the inhibitor or PBS every 3 days for 6 weeks, and then we examined cardiac function in each group by ultrasonic

echocardiography at various time points (Supplementary Fig. 5k, l). Left ventricular dilatation with reduced fractional shortening was partially but significantly suppressed in the ATM inhibitor-treated group relative to the PBS control group (Supplementary Fig. 5l and Supplementary Table 1). Furthermore, we observed no difference in lung weight/body weight ratio between groups, while the heart weight/body weight ratio was significantly decreased in inhibitor-treated relative to the PBS group (Supplementary Fig. 5m). ATM activation was also markedly inhibited in the heart of the inhibitor-treated relative to the PBS group (Supplementary Fig. 5n). These results suggest that enhanced ATM activation partially contributes to cardiac dysfunction in TAC-operated *Caren* ^{β -geo/ β -geo} mice. Because *Caren* ^{β -geo/ β -geo} mice showed decreased mitochondrial respiratory capacity in the heart, both enhanced ATM-DDR pathway activation and mitochondrial dysfunction in cardiomyocytes may accelerate cardiac dysfunction and shorten survival of *Caren* ^{β -geo/ β -geo} mice after TAC surgery. We report these findings in the Results section on Page 16, Lines 8–20.

Since ATM inhibition suppresses HF progression in TAC-operated *Caren* ^{β -geo/ β -geo} mice, in the future, we will assess whether ATM inhibitor treatment prolongs survival of *Caren* ^{β -geo/ β -geo} mice after TAC surgery.

Minor points:

1) *Please define “HF” upon first use in the text.*

We have defined “HF” upon first use in the revision.

2) *Fig. 3j, Fig.6h, Fig. 7l and supplementary Fig.4h. Please add the Western blot for unphosphorylated ATM as a control that Caren does not influence the amount of total ATM. This will support the conclusion that Caren prevents ATM from becoming fully activated due to the suppression of Hint1 translation and not due to any effect that Caren may have on the amount of ATM protein in the cell.*

Based on your comments, we added the data to immunoblotting analysis showing total ATM in order to normalize pATM levels to total ATM.

3) *In the discussion section the authors state that “However, Hint1 reportedly functions as a tumor suppressor by activating ATM and the subsequent DDR.” This statement is not completely correct. Hint1 does not activate ATM, it contributes to the activation of ATM upon DNA damage. So it is the DNA damage that activates ATM and Hint1 helps during this process. I realize that the reason for DNA damage dependent ATM activation upon TAC is not known and well beyond the scope of this paper, but I would please ask the authors to clarify this statement and maybe add a line or two as to how TAC could theoretically cause DNA damage that would activate the DNA damage response kinase ATM.*

Thank you for this clarification. Recent studies suggest that DNA damage is induced in cardiomyocytes during TAC-induced heart failure development (Nakada *et al.*, *Circulation* 139: 1237–1239, 2019; Higo *et al.*, *Nat Commun* 8: 15104, 2017). Increased production of reactive oxygen species (ROS) due to mitochondrial dysfunction is observed in TAC heart and scavenging of ROS reportedly prevents TAC-induced heart failure (Dai *et al.*, *Cardiovasc Res* 93: 79–88, 2012; Peoples *et al.*, *Exp Mol Med* 51: 1–13, 2019). Thus, oxidative stress following excess ROS production from mitochondria underlies TAC-induced DNA damage in cardiomyocytes. We now state this explicitly, as suggested, in the Introduction on Page 3 Line 19, through Page 4, Line 4.

Reviewer #1 (Remarks to the Author):

Questions 1&3: the authors did pulldown for RNA-RNA interaction and concluded that Caren might interact with Hin1 mRNA indirectly. Authors didn't put this data into manuscript and said they will do more experiments in the future. To me, there should be other assays that fit more to study RNA-RNA interaction such as EMSA or RAP (RNA antisense purification). These data are needed.

Reviewer #2 (Remarks to the Author):

The Authors have done a great job addressing my concerns and applaud them for their efforts. With that being said, one issue still remains. With the conclusion being drawn that Caren impacts mitochondrial biogenesis and OXPHOS as a result of an increase in mitochondrial number, the inclusion of some sort of index corroborating this conclusion should be performed. The best example would be EM of sectioned mitochondrial tissue. Other possibilities would be citrate synthase, etc... though this would not be as convincing as the EM.

Reviewer #3 (Remarks to the Author):

I think the authors properly responded to my comment and also to the other reviewers' comments.

Reviewer #4 (Remarks to the Author):

The authors have addressed all my concerns and I can recommend the manuscript be accepted.

Responses to comments by Reviewer 1

Thank you for constructive comments, which were extremely helpful. We have extensively revised the manuscript based on those suggestions and feel that our manuscript is much improved. Our answers are as follows.

Questions 1&3: the authors did pulldown for RNA-RNA interaction and concluded that *Caren* might interact with *Hint1* mRNA indirectly. Authors didn't put this data into manuscript and said they will do more experiments in the future. To me, there should be other assays that fit more to study RNA-RNA interaction such as EMSA or RAP (RNA antisense purification). These data are needed.

Based on your comments, we have added the RNA pull-down assay, which is now shown in Fig. 5f of the revision. In this assay, we reassessed *Caren/Hint1* mRNA interaction using biotinylated *Caren* (2.6 kb) and *LacZ* mRNA (3 kb), the latter as a negative control. When these biotinylated bait RNAs were incubated with cytoplasmic lysates of mouse myoblast C2C12 cells, *Hint1* mRNA abundantly bound to *Caren* relative to *LacZ* mRNA. However, when biotinylated *Caren* transcripts were incubated with total RNA isolated from C2C12 cells, levels of *Hint1* mRNA bound to *Caren* markedly decreased, suggesting that the *Caren/Hint1* interaction is indirect.

Also, based on your suggestions, we performed RAP analysis using mouse embryonal carcinoma P19.CL6 cells to further validate *Caren/Hint1* mRNA interaction. Since the results above suggest it indirectly interacts with *Hint1* mRNA, cells crosslinked with disuccinimidyl glutarate (DSG) and formaldehyde (FA) were subjected to RAP analysis. Moreover, because the *Caren* transcript harbors numerous repeat sequences, it was challenging to design specific probes against *Caren* transcripts. Instead, we carried out RAP using specific probes against *Hint1* mRNA (*Hint1* RAP) and *LacZ* mRNA (*LacZ* RAP, as a negative control) (Fig. 5g). In *Hint1* RAP, average yield of *Hint1* mRNA was approximately 10% and amount of purified *Hint1* mRNA in *Hint1* RAP was more than 10,000-fold higher than that seen in *LacZ* RAP. In *Hint1* RAP analysis, *Caren* transcripts co-precipitated with *Hint1* mRNA, indicative of interaction. By contrast, we did not observe *Caren/Hint1* mRNA interaction in *LacZ* RAP analysis. Furthermore, *Caren/Hint1* mRNA interaction significantly decreased in *Hint1* RAP analysis using non-crosslinked cells. We now report these findings in the Results section on Page 13, Line 20, through Page 14, Line 17.

Responses to comments by Reviewer 2

Thank you for constructive comments, which were extremely helpful. We have extensively revised the manuscript based on those suggestions and feel that our manuscript is much improved. Our answers are as follows.

The Authors have done a great job addressing my concerns and applaud them for their efforts. With that being said, one issue still remains. With the conclusion being drawn that *Caren* impacts mitochondrial biogenesis and OXPHOS as a result of an increase in mitochondrial number, the inclusion of some sort of index corroborating this conclusion should be performed. The best example would be EM of sectioned mitochondrial tissue. Other possibilities would be citrate synthase, etc... though this would not be as convincing as the EM.

Thank you for your comment. Accordingly, we have now assessed the number of mitochondria in heart tissue of *CAG-Caren* Tg and WT littermate mice using EM analysis. That analysis revealed that the number of mitochondria in heart tissue of *Caren* Tg mice is greater than the number seen in WT mice (Fig. 4h). We now report these findings in the Results section on Page 11, Lines 7–9.

Reviewer #1 (Remarks to the Author):

No further comments.

Reviewer #2 (Remarks to the Author):

No further comments.